# Generalized Bézier-like model and its applications to curve and surface modeling

**Moavia Ameer[1], Muhammad Abbas ⓘ[1], Madiha Shafiq[1], Tahir Nazir[1], Asnake Birhanu ⓘ[2]\***

1 Department of Mathematics, University of Sargodha, Sargodha, Pakistan, 2 Department of Mathematics, College of Science, Hawassa University, Hawassa, Ethiopia

\* asnakeb@hu.edu.et

**Data Availability Statement:** All relevant data are within the paper.

**Funding:** The author(s) received no specific funding for this work;.

## Abstract

The subject matter of surfaces in computer aided geometric design (CAGD) is the depiction and design of surfaces in the computer graphics arena. Due to their geometric features, modeling of Bézier curves and surfaces with their shape parameters is the most well-liked topic of research in CAGD/computer-aided manufacturing (CAM). The primary challenges in industries such as automotive, shipbuilding, and aerospace are the design of complex surfaces. In order to address this issue, the continuity constraints between surfaces are utilized to generate complex surfaces. The parametric and geometric continuities are the two metrics commonly used for establishing connections among surfaces. This paper proposes continuity constraints between two generalized Bézier-like surfaces (gBS) with different shape parameters to address the issue of modeling and designing surfaces. Initially, the generalized form of $C^3$ and $G^3$ of generalized Bézier-like curves (gBC) are developed. To check the validity of these constraints, some numerical examples are also analyzed with graphical representations. Furthermore, for a continuous connection among these gBS, the necessary and sufficient $G^1$ and $G^2$ continuity constraints are also developed. It is shown through the use of several geometric designs of gBS that the recommended basis can resolve the shape and position adjustment problems associated with Bézier surfaces more effectively than any other basis. As a result, the proposed scheme not only incorporates all of the geometric features of curve design schemes but also improves upon their faults, which are typically encountered in engineering. Mainly, by changing the values of shape parameters, we can alter the shape of the curve by our choice which is not present in the standard Bézier model. This is the main drawback of traditional Bézier model.

## 1 Introduction

Bézier curves have a lot of applications in the fields of science, engineering, and technology such as: railway route and highway modeling, network, computer aided design system, robotics, environment design, communication and many other fields just because of their computational simplicity and stability. In addition to being significant research tools in CAGD, parametric curves and surfaces are also effective tools for shape design and geometric interpretation [1, 2]. In a real-world problems, most of the curves and surfaces are complex to model.

**Competing interests:** The authors have declared that no competing interests exist.

Therefore, having a tool to create flexible and adjustable curves and surfaces becomes necessary to model any complex shapes. However, finding a representation for the desired complex curves and surfaces is difficult and impractical [3]. To tackle these problems, the curves and surfaces are broken down into simpler curve and patches, joining them to form of complex curves or surfaces. To join such curves or surfaces, a continuity concept is applied. Smoother connections between curves or surfaces typically indicate a higher level of continuity. Two standard functions have been used in connecting these curves or surfaces; parametric and geometric continuity. The parametric continuity of order $r$, or $C^r$ is the simplest form of the continuity. Nonetheless, parametric continuity has some restrictions. For example, when the surfaces are connected by $C^1$ continuity, they still need to possess a common tangent at their boundary points. Hence, parametric continuity cannot be the only exact standard method in constructing smooth curves or surfaces. As a result, the researchers have developed an upgraded version of parametric continuity called geometric continuity, or $G^r$ continuity. $G^r$ continuity is the less restrictive form where scale factors are embedded in the continuity. These scale factors overcome a common tangent for the curves or surfaces. The classical Bézier curves have some flaws and limitations due to their fixed shape and position relative to their control polygon. A lot of work has been carried out to overcome these flaws. The shape and positions of the curve is enhanced by introducing the shape control parameters in the Bézier approach. The shape of the curve is control by utilizing the different values of shape parameters.

Bernstein basis functions are often used to create classical Bézier curves and surfaces because they have a clear definition and a number of advantageous features. As a result, Bézier modeling has already established itself as one of the crucial techniques for describing complex curves and surfaces in the CAGD area [4, 5]. De Casteljau became the leading person to define the Bézier surfaces by using triangular domain in the late 1950s. Boehm carried on his research, and his achievements are documented in [6]. Throughout the 1970s and 1980s, different researchers did comprehensive research on the Bézier triangular surface [7–11]. Meanwhile, Abedallah [12] worked on the Bézier triangular logic surface. This surface theory was improved through the participation of such philosophers. Numerous researchers are fascinated by the triangular surface modeling framework only because of how well it may construct complicated structures. However, only a few prior works have focused on refining the triangular surface to enhance the traditional triangular Bézier structure after the Bézier-like surfaces theory has fully matured [13–16].

A revised basis function with $n - 1$ shape parameters was provided by Qin *et al.* [17]. As an application, they developed countless surfaces and curves. The biquintic Bézier surfaces were developed by Ammad *et al.* [18]. They discussed the geometric characteristics of surfaces and offered some surfaces with shape customization. The Bézier-like surfaces with distinct shape parameters that have similar geometric characteristics to BC, were first introduced by Hu *et al.* [19]. They established the necessary and sufficient conditions for $G^1$ continuity, $G^2$ beta continuity and Farin-Boehm $G^2$ continuity across two successive developed surfaces. Hu *et al.* [20] proposed the generalized Bézier surface with various shape parameters and highlighted its applications in engineering. Li *et al.* [21] generalized the H-Bézier model and derived the necessary and sufficient conditions of first and second order geometric continuity. The $G^1$ geometric continuity constraints of two successive H-Bézier surfaces are also illustrated. In addition, certain H-Bézier surface modeling examples are shown to demonstrate the effectiveness of computer design for complex curves and surface models. Bibi *et al.* [22] used graphical depiction to construct the necessary and sufficient $G^2$ continuity requirements of two sequential generalized hybrid trigonometric Bézier surfaces in various directions. They also formed some free-form complex engineering surfaces. Hu *et al.* [23] modeled the gBS with

independent shape parameters. They gave conditions for $G^2$ continuity between two adjacent gBS. They also discussed some properties and applications of the smooth continuity between these surfaces. Mad *et al.* [24] discussed the fractional continuity of degree two (or $F^2$) for generalized fractional Bézier surfaces, which can be used to alter the shape without changing the control points.

This work presents the new formulation for the surfaces by utilizing the gBS. The generalized Bernstein-like basis function has two different shape parameters which help to alter the shape without changing the value of control points. The generalized form of parametric $C^3$ and geometric $G^3$ continuity constraints are constructed and some numerical examples are presented to check the validity of these constraints. The continuity constraints of surfaces are presented by using the gBS and their applications are also given in this study. This paper's contribution is the continuity extension of [25, 26] carried out by Ameer *et al*. The following are some of the contributions made by this work:

- The generalized form of parametric $C^3$ and geometric $G^3$ continuity constraints are constructed.

- The effect of shape parameters are analyzed by some numerical examples.

- The generalized form of $G^2$ Bézier-like surfaces continuity constraints with two distinct parameters are constructed.

- To check the validity of these continuity constraints, some figures are formed.

The following is an outline of the article: Section 2 contains preliminary details on curvature, generalized Bernstein-like basis functions and gBC with characteristics, as well as constraints on parametric and geometric continuity. In section 3, we construct surface continuity requirements for gBS with two shape parameters. Section 4 provides the work's conclusion.

## 2 Preliminaries

When constructing a curve of any type, the cartesian coordinate system is often used for the majority of the work. Because of this, the control points are considered to exist in two tuples concurrently.

In a similar vein, both points and vectors are presented in this article in boldface throughout its entirety. e.g.,

$$U = \begin{pmatrix} u \\ v \end{pmatrix} \tag{1}$$

and the following equation can be used to define the Euclidean norm of a vector $||\mathbf{U}(\Phi)|| = \sqrt{u^2 + v^2}$. The notation for the derivative of the function $U(\Phi)$ is written as $U'(\Phi)$.

### 2.1 Curvature

The term "curvature" refers to one of the most essential ideas in geometry, which is a subfield of mathematics. The fundamental idea behind the term curvature refers to the extent to which a curve veers away from the position of a straight line by a certain amount. For any parametric curve denoted by the symbol "$U(\Phi)$," the curvature, in addition to its corresponding

expression in mathematics, can be described as follows:

$$\kappa(\Phi) = \frac{U'(\Phi) \times U''(\Phi)}{||U'(\Phi)||^3}. \tag{2}$$

For any parametric curve that exists in two dimensions, the curvature can alternatively be stated as follows:

$$\kappa(\Phi) = \frac{u'v'' - v'u''}{\left[(u')^2 + (v')^2\right]^{\frac{3}{2}}}, \tag{3}$$

where $u$ and $v$ are the x and y components of $U(\Phi)$. In summary, the definition of the rate of change of curvature, symbolised by the symbol $\kappa'(\Phi)$, is obtained by calculating the first derivative of Eq (2) as:

$$\kappa'(\Phi) = \frac{||U'(\Phi)||^2 \{U'(\Phi) \times U'''(\Phi)\} - 3\{U'(\Phi) \times U''(\Phi)\}\{U'(\Phi) \cdot U''(\Phi)\}}{||U'(\Phi)||^5}, \tag{4}$$

where $||U'(\Phi)||$ is the magnitude of the tangent vector. As a consequence, the reciprocal of the curvature can be used to determine the radius of curvature. By taking the first derivative of Eq (3), we obtain:

$$\kappa'(\Phi) = \frac{(u'^2 + v'^2)(u'v''' - v'u''') - 3(u'v'' - v'u'')(u'u'' + v'v'')}{\left[(u')^2 + (v')^2\right]^{\frac{5}{2}}}. \tag{5}$$

## 2.2 Generalized Bernstein-like functions

This section discusses the definition of generalized Bernstein-like (gB-like) functions as well as their properties, specifically those functions having two distinct shape parameters.

**Definition 1** *Given $\varsigma, \zeta \in [0, 3]$, for $\tilde{w} \in [0, 1]$, the functions*

$$\begin{cases} \breve{u}_{0,2}(\tilde{w}) = (1 - \tilde{w})^2 (1 + (2 - \varsigma)\tilde{w}), \\ \breve{u}_{1,2}(\tilde{w}) = \tilde{w}(1 - \tilde{w})(\varsigma + \tilde{w}(\zeta - \varsigma)), \\ \breve{u}_{2,2}(\tilde{w}) = \tilde{w}^2(3 - \zeta + \tilde{w}(\zeta - 2)), \end{cases} \tag{6}$$

*are known as quadratic Bernstein-like functions* [25].

For any integer $r(r \geq 3)$, the functions $\breve{u}_{f,r}(\tilde{w})(f = 0, 1, \ldots, r)$ are recursively explained by:

$$\breve{u}_{f,r}(\tilde{w}) = (1 - \tilde{w})\breve{u}_{f,r-1}(\tilde{w}) + \tilde{w}\breve{u}_{f-1,r-1}(\tilde{w}), \quad \tilde{w} \in [0, 1], \tag{7}$$

where $\breve{u}_{f,r}(\tilde{w})$ are known as gB-like functions of degree $r$ [25]. These functions become zero when $f < 0$ or $f > r$. The gB-like basis functions are stated in a precise manner as follows:

$$\begin{aligned} \breve{u}_{f,r} &= \left[ \frac{{}^rC_f - 2({}^{r-3}C_{f-1} - {}^{r-3}C_{f-3})}{{}^rC_f} + \frac{{}^{r-2}C_{f-1}}{{}^rC_f}\varsigma - \frac{{}^{r-2}C_{f-2}}{{}^rC_f}\zeta - \frac{{}^{r-1}C_f}{{}^rC_f}\varsigma\tilde{w} \right. \\ &\quad \left. + \frac{{}^{r-1}C_{f-1}}{{}^rC_f}\zeta\tilde{w} + 2\tilde{w}\frac{({}^{r-2}C_f - {}^{r-2}C_{f-2})}{{}^rC_f} \right]{}^rC_f\tilde{w}^f(1 - \tilde{w})^{r-l}, \end{aligned} \tag{8}$$

where $rC_f = \frac{r!}{f!(r-f)!}$ and $f = 0, 1, \cdots, r, r \geq 3$. In [25], the geometric characteristics of gB-like basis functions are shown to be proven.

## 2.3 Generalized Bézier-like curves and surfaces

The following section covers the definition of Bézier-like curves and surfaces with two shape parameters.

## 2.4 Construction of the generalized Bézier-like curve with two shape parameters

**Definition 2** *Given control points* $D_f \in R^k (k = 2, 3; f = 0, 1, \cdots, r; r \geq 2)$, *we call*

$$W(\tilde{w}) = \sum_{f=0}^{r} \breve{u}_{f,r}(\tilde{w}) D_f, \qquad \tilde{w} \in [0, 1], \tag{9}$$

*the Bézier-like curve of order r, where* $\breve{u}_{f,r}(\tilde{w})(f = 0, 1, \ldots, r)$ *are the Bernstein-like basis functions.*

The geometric properties of gBC are proved in [25].

Fig 1 depicts the cubic Bézier-like curve for the shape parameter values $\varsigma = 2$ and $\zeta = 2$. Fig 2 shows the cubic Bézier-like curve for the same values of shape parameters. Fig 3 is constructed by different values of shape parameters, which shows that with these values of shape parameters, the curves get closer. The Figs 2 and 3 show the effect of shape parameters on the curve and modification in curve by changing the values of shape parameters. Note that if the value of shape parameter is large then the curve is close to the control net and by decreasing the value of shape parameter it is going away from the control net.

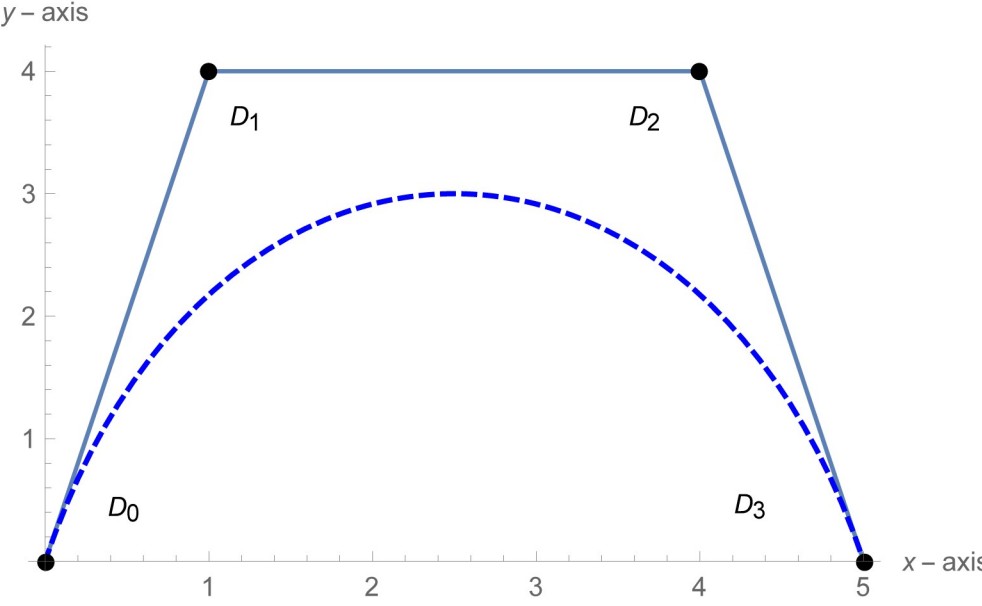

**Fig 1. Cubic Bézier-like curve for** $(\varsigma, \zeta) = (2,2)$.

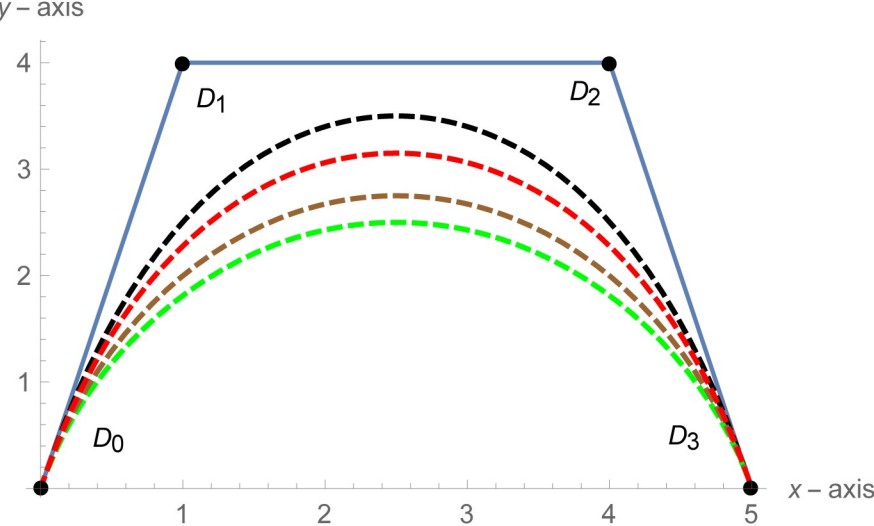

**Fig 2. Cubic Bézier-like curve for same values of shape parameters, Green:** $(\varsigma, \zeta) = (1,1)$, **Brown:** $(\varsigma, \zeta) = (1.5, 1.5)$, **Red:** $(\varsigma, \zeta) = (2.3, 2.3)$, **Black:** $(\varsigma, \zeta) = (3,3)$.

## 2.5 Establishment of Bézier-like surfaces with two shape parameters

**Definition 3** *For the control points array $D_{fg} \in R^3$, where $(f = 0, 1, 2, \cdots, r)$, $(g = 0, 1, 2, \cdots, s)$ and $f, g \geq 3$, the tensor product is specified as:*

$$W(\tilde{w}, \tilde{w}1; \zeta, \varsigma, \zeta1, \varsigma1) = \sum_{f=0}^{r}\sum_{g=0}^{s} D_{f.g}\breve{u}_{f,r}(\tilde{w})\breve{u}_{g,s}(\tilde{w}1) \qquad 0 \leq \tilde{w}, \tilde{w}1 \leq 1, \qquad (10)$$

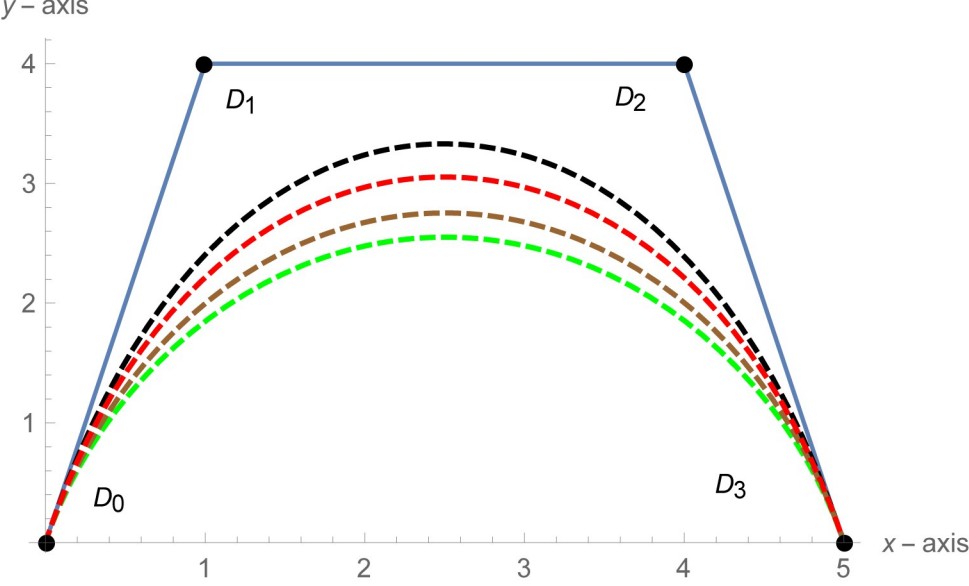

**Fig 3. Cubic Bézier-like curve for different values of shape parameters, Green:** $(\varsigma, \zeta) = (1, 1.2)$, **Brown:** $(\varsigma, \zeta) = (1.3, 1.5)$, **Red:** $(\varsigma, \zeta) = (1.7, 1.8)$, **Black:** $(\varsigma, \zeta) = (2.2, 2.5)$.

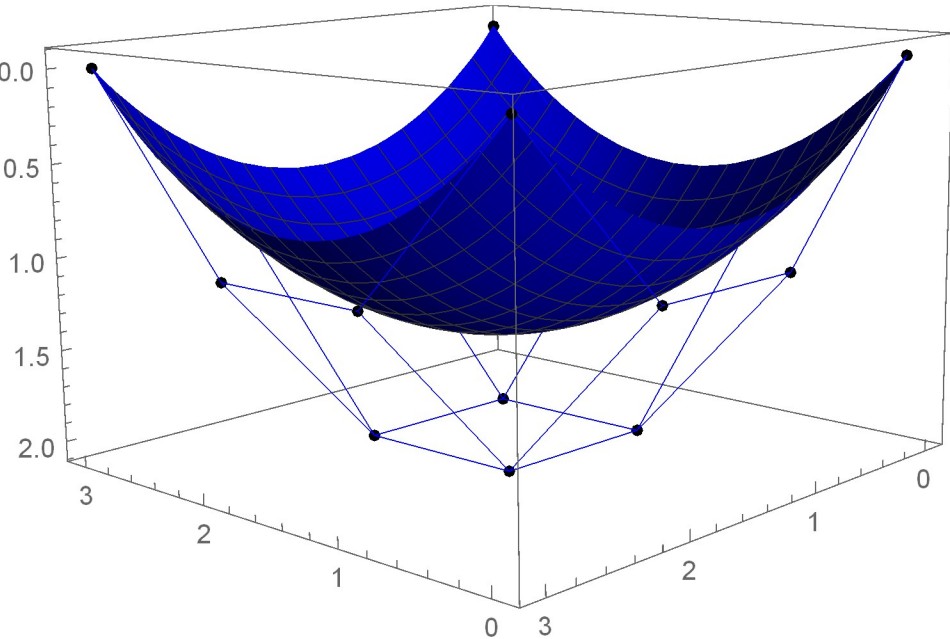

**Fig 4. Rectangular Bézier-like surfaces with the same control net for different values of shape parameters** $\varsigma = 1$, $\zeta$ **= 2 and** $\varsigma 1 = 2$, $\zeta 1 = 2$.

which is called the Bézier-like order surface $r{\times}s$ with control points $W_{f,g}$, where $\breve{u}_{f,r}(\tilde{w})$ and $\breve{u}_{g,r-1}(\tilde{w1})$ are Bernstein-like basis functions. The $\varsigma$, $\zeta$ and $\varsigma 1$, $\zeta 1$ are the shape parameters for the basis functions $\breve{u}_{f,r}(\tilde{w})$ and $\breve{u}_{g,s}(\tilde{w1})$, respectively.

Similarly, we can construct the Bézier-like surfaces and alter the shape of surfaces by our choice. To check the effect of shape parameters on the surfaces, Figs 4–6 are constructed by different values of shape parameters, which shows that by changing the value of shape parameters the shape of the surfaces are adjusted. As the value of the shape parameters increases the surface becomes closer to the control net, which is helpful to construct the different models in engineering and many other fields.

### 2.6 Continuity constraints of Bézier-like curves

In this part, we will discuss about the parametric and geometric constraints of Bézier-like curves. When it comes to making complicated figures and doing different kinds of modeling, the continuity conditions are particularly helpful. Here, we discuss the $C^3$ and $G^3$ continuity of Bézier-like curves. The aim of reaching $C^3$ and $G^3$ continuity after $C^2$ and $G^2$ continuity, is to achieve a high level of smoothness in order to make intricate figures. Consequently, the continuity of $C^3$ and $G^3$ must initially contain lower order continuity.

The next theorem is for the generalization of $C^3$ parametric continuity constraints, the rest has been established in Ameer *et al.* [26].

**Theorem 1** *Consider any two Bézier-like curves* $W(\tilde{w}) = \sum\limits_{f=0}^{r} D_f \breve{u}_{f,r}(\tilde{w})$ *with control points as*

$D_0, D_1, \ldots, D_r$, *where* $r \geq 3$ *and* $W_1(\tilde{w}) = \sum\limits_{g=0}^{s} E_g \breve{u1}_{g,s}(\tilde{w})$ *with control points* $E_0, E_1, \ldots, E_s$,

$s \geq 3$, *these curves meet to the parametric continuity constraints if and only if (iff):*

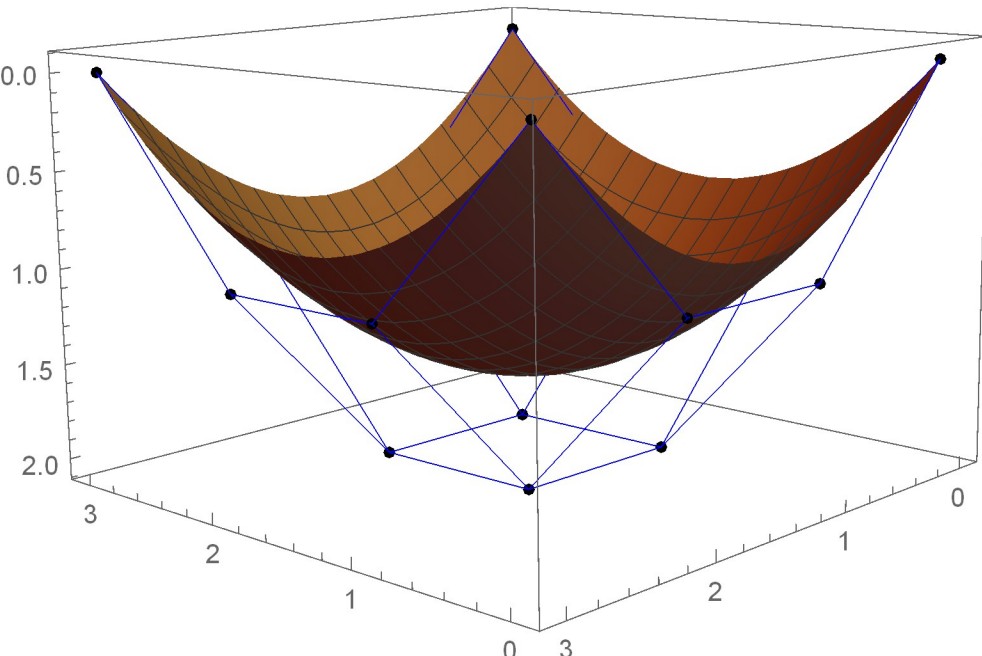

**Fig 5. Rectangular Bézier-like surfaces with the same control net for different values of shape parameters** $\varsigma = 2.5$, $\zeta = 2.5$ **and** $\varsigma 1 = 2$, $\zeta 1 = 2$.

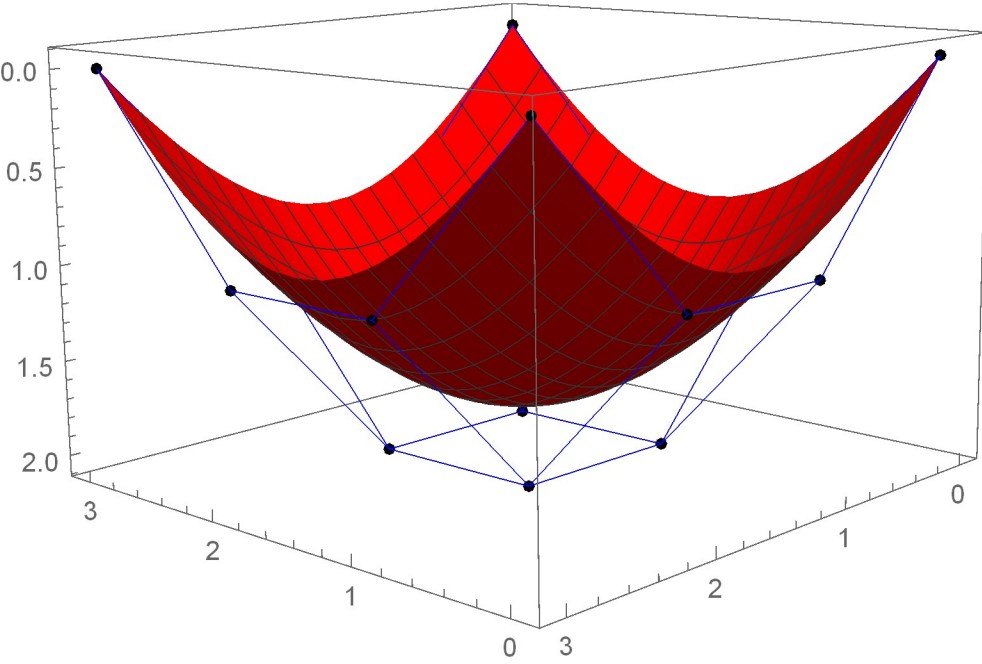

**Fig 6. Rectangular Bézier-like surfaces with the same control net for different values of shape parameters** $\varsigma = 3$, $\zeta = 3$ **and** $\varsigma 1 = 3$, $\zeta 1 = 3$.

*1. For $C^0$ continuity:*

$$D_r = E_0. \tag{11}$$

*2. For $C^1$ continuity:*

$$\begin{cases} D_r = E_0, \\ \\ E_1 = D_r + \dfrac{(r - 2 + \zeta)}{(s - 2 + \varsigma_1)}(D_r - D_{r-1}), \end{cases} \tag{12}$$

*3. For $C^2$ continuity:*

$$\begin{cases} D_r = E_0, \\ E_1 = D_r + \frac{(r-2+\zeta)}{(s-2+\varsigma_1)}(D_r - D_{r-1}), \\ E_2 = \frac{1}{(s-2+\varsigma_1)(s+^{s-3}C_2+(s-2)\varsigma_1-\zeta_1)}[(r+^{r-3}C_2 - \varsigma + (r-2)\zeta)(s-2+\varsigma_1)D_{r-2} \\ +(D_r - D_{r-1})[(s-2)(s-2)(s+r-6) + (s-2)(r+s-3)\zeta + (2(r-2)s+ \\ (r-2)(r-5))\varsigma_1 + (r+2s-2)\zeta\varsigma_1 - (r-2)\zeta_1 - \zeta\zeta_1] \\ +V_{r-1}(\varsigma - (r-2)\zeta)(r-2+\varsigma_1) + D_r(\frac{(s-2)(s-r)(s+r-5)}{2} + (s-2)\varsigma_1^2 \\ -(s-2)\zeta_1 - \varsigma_1\zeta_1 + \frac{1}{2}((3-r)(r-2) + (s-2)(3s-7))\varsigma_1)]. \end{cases} \tag{13}$$

*4. For $C^3$ continuity constraints, we used the constraints of $C^2$ continuity and obtained the new fourth control points $E_3$ in $C^3$ continuity. As a consequence, we have:*

$$\begin{aligned} E_3 = & -\tfrac{1}{12+(-5+s)s+2(-2+s)\varsigma_1-2\zeta_1}[(-r(-5+r+2\zeta) \\ & +2(-6+\varsigma+2\zeta))D_{r-2} - 2(-6+\varsigma+2\zeta) - r(-5+r+2\zeta)D_{r-1} \\ & -r(-5+r+2\zeta)D_r + sD_r(-5+s+2\varsigma_1) + 2(-6-(-5+s)s \\ & +\zeta - 2(-1+s)\varsigma_1)\Big(D_r + \tfrac{(-2+r+\zeta)(D_r-D_{r-1})}{-2+\varsigma_1+s}\Big)]. \end{aligned} \tag{14}$$

**Proof** The $C^f$ ($f$ = 0, 1, 2) continuity is already proved in [26]. The $C^3$ continuity can be achieved by using the $C^2$ continuity conditions and $W'''(1) = W_1'''(0)$. After some simplifications, we can obtain the $C^3$ continuity conditions which is given in Eq (14).

The next theorem is for the generalization of $G^3$ geometric continuity constraints, the rest has been established in Ameer *et al.* [26].

**Theorem 2** *Consider any two Bézier-like curves $W(\tilde{w}) = \sum_{f=0}^{r} D_f \breve{u}_{f,r}(\tilde{w})$ with control points $D_0, D_1, \ldots, D_r$, where $r \geq 3$ and $W_1(\tilde{w}) = \sum_{g=0}^{s} E_l \breve{u} 1_{g,s}(\tilde{w})$ with control points $E_0, E_1, \ldots, E_s$, $s \geq 3$, these curves meet to the geometric continuity constraints if and only if:*

*i. $D_r = E_0$ for $G^0$ continuity.*

*ii. For $G^1$ continuity:*

$$D_r = E_0, E_1 = D_r + \frac{(r - 2 + \zeta)}{\varsigma(s - 2 + \varsigma_1)}(D_r - D_{r-1}),$$

*when ($\varsigma = 1$, the continuity of $G^1$ becomes the continuity of $C^1$.*

*iii. For $G^2$ continuity, $r \geq 3, s \geq 3$*

$$
\begin{aligned}
E_2 = \frac{1}{2\varsigma^2(s - 2 + \varsigma_1)(s +^{s-3}C_2 + (s-2)\varsigma_1 - \zeta_1)}&[2(r +^{r-3}C_2 - \varsigma + (s-2)\zeta)(s \\
-2 + \varsigma_1)E_{r-2} + (E_r - E_{r-1})&[(s-2)(-1 + 2(s-3)\varsigma)(r - 2 + \zeta) + r(2s-4)\zeta \\
+2r\zeta\varsigma_1 + \varsigma_1(-1 + (4s-4)\varsigma)(r - 2 + \zeta) + D_{r-1}&2(s - 2 + \varsigma_1)(\varsigma - (r-2)(w + \zeta \\
-3)) + D_r\left[2\left(\frac{r(r-5)(r-2)}{2} - 2\varsigma(r - 2 + \zeta)\zeta_1\right)\right. &\left.+ (s-2)\left(s + \frac{(s-3)(s-4)}{2}\right)\varsigma^2\right) \\
+2\varsigma_1\left(\frac{r(r-5)}{2} + \left(s(s-2) + \frac{(s-5)(s-4)}{2}\right)\varsigma^2\right)& + 2\varsigma^2((s-2)\varsigma_1^2 - (s-2)\zeta_1 - \varsigma_1\zeta_1)]].
\end{aligned}
\tag{15}
$$

*4. For $G^3$ continuity constraints, we used $G^2$ continuity constraints and also obtained the new fourth control points $E_3$ in $G^3$ continuity. Therefore, we have:*

$$
\begin{aligned}
E_3 = -\frac{1}{\chi}\big[&-(2-r)(30 - 6\varsigma - 9\zeta + r(-7 + r + 3\zeta))D_{i,-3+r} - 3(-32 + 6\varsigma \\
&+10\zeta + r(32 - 4\varsigma - 11\zeta + r(-9 + r + 3\zeta)))D_{i,-2+r} + 3(-1 + r)(-2(-6 + \varsigma \\
&+2\zeta) + r(-8 + r + 3\zeta))D_{i,-1+r} - (-1 + r)r(-8 + r + 3\zeta)D_{i,r} + (1 - s)s(-8 \\
&+s + 3\varsigma1)\psi D_{i,r} + \chi_1 + 3(-1 + s)(s(-8 + s + 3\varsigma1) - 2(-6 + 2\varsigma1 + \zeta1))\psi(D_{i,r} \\
&+\chi_2) - \chi_3(3(-32 + 10\varsigma1 + s(32 - 11\varsigma1 + s(-9 + s + 3\varsigma1) - 4\zeta1) \\
&+6\zeta1)\psi(2(-2 + s + \varsigma1)(\tfrac{1}{2}(-4 + r)(-3 + r) + r - \varsigma + (-2 + r) + \zeta)D_{i,-2+r} \\
&+2(-2 + s + \varsigma1)(\varsigma - (-2 + r)(-3 + r + \zeta))D_{i,r-1} + 2(\tfrac{1}{2}(-5 + r)(-2 + r)r \\
&+(-2 + s)(\tfrac{1}{2}(-4 + s)(-3 + s) + s)\alpha^2 + 2(\tfrac{1}{2}(-5 + r)r + (\tfrac{1}{2}(-5 + s)(-4 + s) \\
&+(-2 + s)s)\alpha^2)\varsigma1) + 2\alpha^2((-2 + s)(\varsigma1)^2 - (-2 + s)\zeta1 - \varsigma1\zeta1))D_{i,r} + ((-4 \\
&+2s)r\zeta + 2r\varsigma1\zeta + (-2 + s)(-1 + 2(-3 + s)\alpha)(-2 + r + \zeta) + (-1 + (-4 \\
&+4s)\alpha)\varsigma1(-2 + r + \zeta) - 2\alpha(-2 + r + \zeta)\zeta1)(-D_{i,-1+r} + D_{i,r})) + 3\zeta(s(-5 \\
&+s + 2\varsigma1)D_{i,r} + 2(-6 + 2\varsigma1 - s(-5 + s + 2\varsigma1) + \zeta)(D_{i,r} + \chi_4) + \chi_5(s(-5 \\
&+s + 2\varsigma1) - 2(-6 + 2\varsigma1 + 2\zeta1))(2(-2 + s + \varsigma1)(\tfrac{1}{2}(-4 + r)(-3 + r) + r - \varsigma \\
&+(-2 + r)\zeta)D_{i,-2+r} + 2(-2 + s + \varsigma1)(\varsigma - (-2 + r)(-3 + r + \zeta))D_{i,-1+r} \\
&+(2(\tfrac{1}{2}(-5 + r)(-2 + r)r + (-2 + s)(\tfrac{1}{2}(-4 + s)(-3 + s) + s)\alpha^2 + 2(\tfrac{1}{2}(-5 \\
&+r)r + (\tfrac{1}{2}(-5 + s)(-4 + s) + (-2 + s)s)\alpha^2)\varsigma1) + 2\alpha^2((-2 + s)\varsigma1^2 - (-2 \\
&+s)\zeta1 - \varsigma1\zeta1))D_{i,r} + ((-4 + 2s)r\zeta + 2r\varsigma1\zeta + (-2 + s)(-1 + 2(-3 \\
&+s)\alpha)(-2 + r + \zeta) + (-1 + (-4 + 4s)\alpha)\varsigma1(-2 + r + \zeta) - 2\alpha(-2 + r \\
&+\zeta)\zeta1)(-D_{i,-1+r} + D_{i,r})))].
\end{aligned}
\tag{16}
$$

*where*

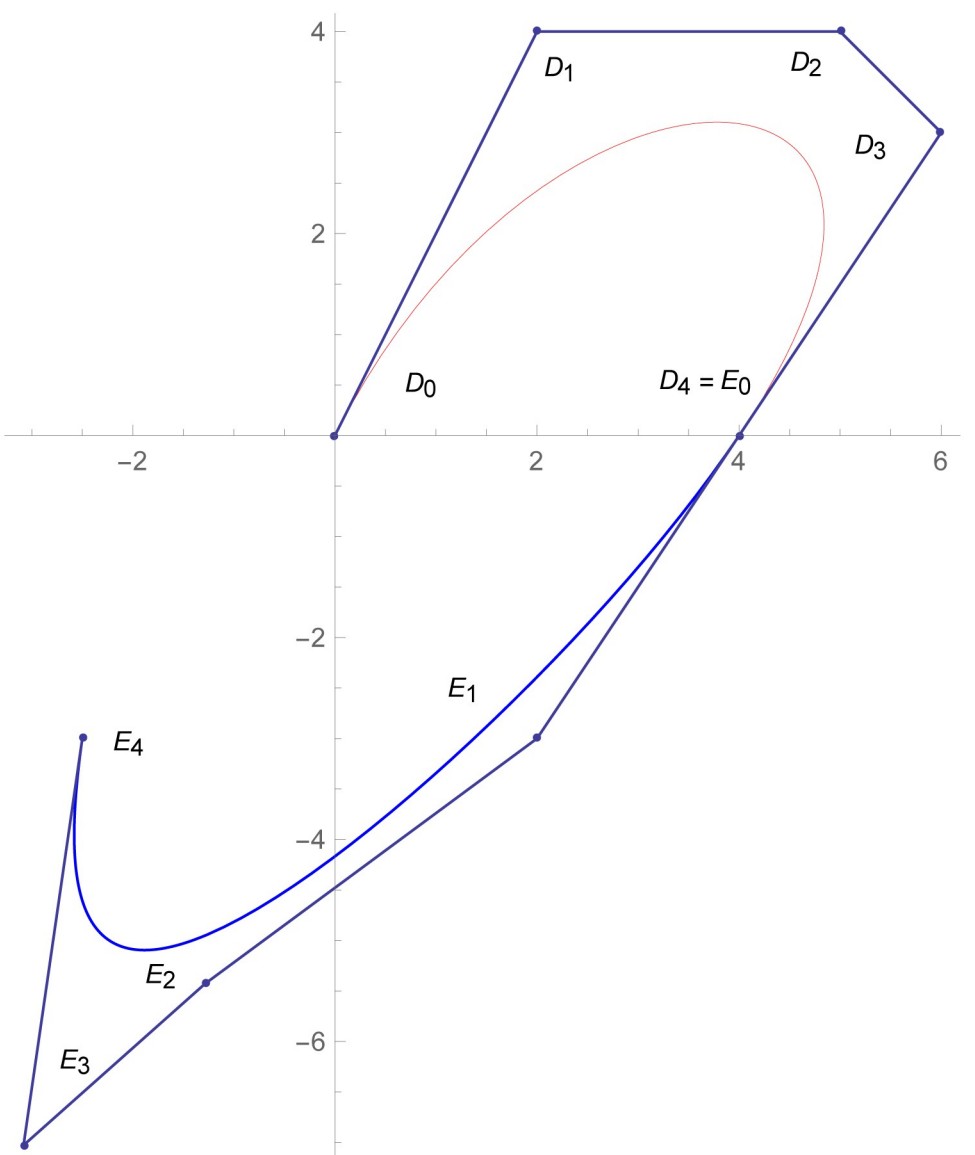

**Fig 7. Quantic Bézier-like curve segments connected by parametric $C^3$ continuity.**

$$\chi = (-2 + s)(30 - 9\varsigma 1 + s(-7 + s + 3\varsigma 1) - 6\zeta 1)\psi,$$

$$\chi_1 = \frac{((-2+r+\varsigma)(-2+r+\zeta)(D_{i,-1+r}+D_{i,r}))}{(-2+s+\varsigma 1)},$$

$$\chi_2 = \frac{(-2+r+\zeta)(-D_{i,-1+r}+D_{i,r})}{\alpha(-2+s+\varsigma 1)},$$

$$\chi_3 = \frac{1}{2\alpha^2(-2+s+\varsigma 1)\left(\frac{1}{2}(-4+s)(-3+s)+s+(-2+s)\varsigma 1-\zeta 1\right)},$$

$$\chi_4 = \frac{(-2+r+\zeta)(D_{i,-1+r}+D_{i,r})}{\alpha(-2+s+\varsigma 1)},$$

*and*

$$\chi_5 = \frac{1}{2\alpha^2(-2+s+\varsigma 1)\left(\frac{1}{2}(-4+s)(-3+s)+s+(-2+s)\varsigma 1-\zeta 1\right)}.$$

**Proof** The $G^k$ ($k = 0, 1, 2$) continuity is already proved in [26]. In order to satisfy the $G^3$ continuity requirements, it is necessary that the first derivative of the curvature at end points must

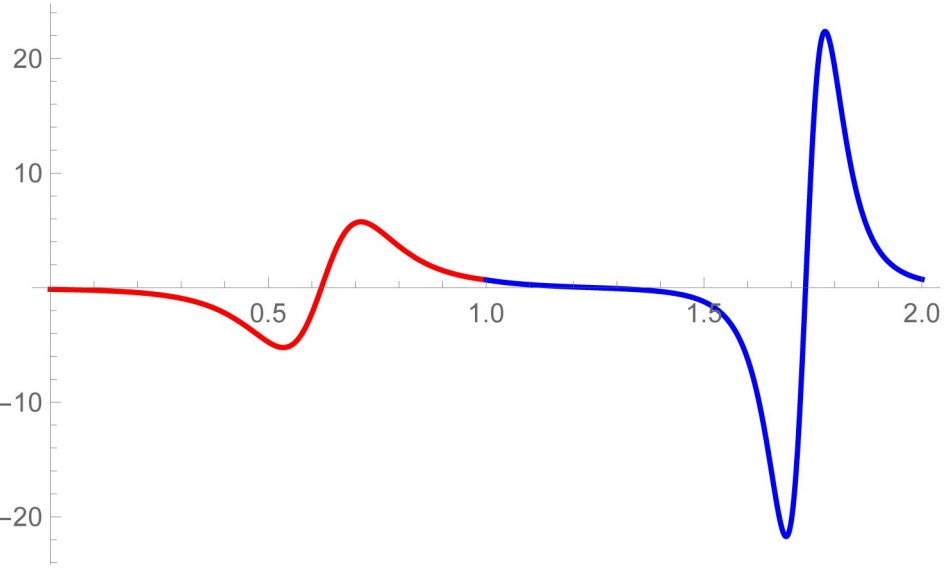

**Fig 8. Curvature plot of quantic Bézier-like curve segments.**

be same as the second derivative at the initial point, i.e., $\kappa_1'(1) = \phi\kappa_2'(0)$. The mathematical expression shows that the first derivative of curvature gives the rate of change of curvature:

$$\kappa_1'(1) = \frac{||W'(1)||^2\{W'(1) \times W'''(1)\} - 3\{W'(0) \times W''(1)\}\{W'(1) \cdot W''(1)\}}{||W'(1)||^5} \qquad (17)$$

and

$$\kappa_2'(0) = \frac{||W'(0)||^2\{W'(0) \times W'''(0)\} - 3\{W'(0) \times W''(0)\}\{W'(0) \cdot W''(0)\}}{||W'(0)||^5}. \qquad (18)$$

The provided conditions are considered as follows:

$$\begin{cases} W_1'(1) = \phi W_2'(0), \\ W_1''(1) = \xi W_2''(0) + \phi w_2'(0), \\ W_1'''(1) = \psi W_2'''(0) + 3\phi^2 W_2''(0) + \phi W_2'(0), \end{cases} \qquad (19)$$

where $\psi = \phi^3$ and $\xi = \phi^2$ meet the $G^3$ continuity constraints. Consider,

$$\kappa_1'(1) = \frac{\varphi}{||\phi^5 W_1'(0)||^5}, \qquad (20)$$

where

$$\begin{aligned} \varphi \quad = \quad & ||\phi W_2'(0)||^2\{\phi W_2'(0) \times (\psi W_2'''(0) + 3\phi^2 W_2''(0) + \phi W_2'(0))\} \\ & -3\{\phi W_2'(0) \times (\xi W_2''(0) + \phi W_2'(0))\}\{\phi W_2'(0) \cdot (\xi W_2''(0) + \phi W_2'(0))\}. \end{aligned}$$

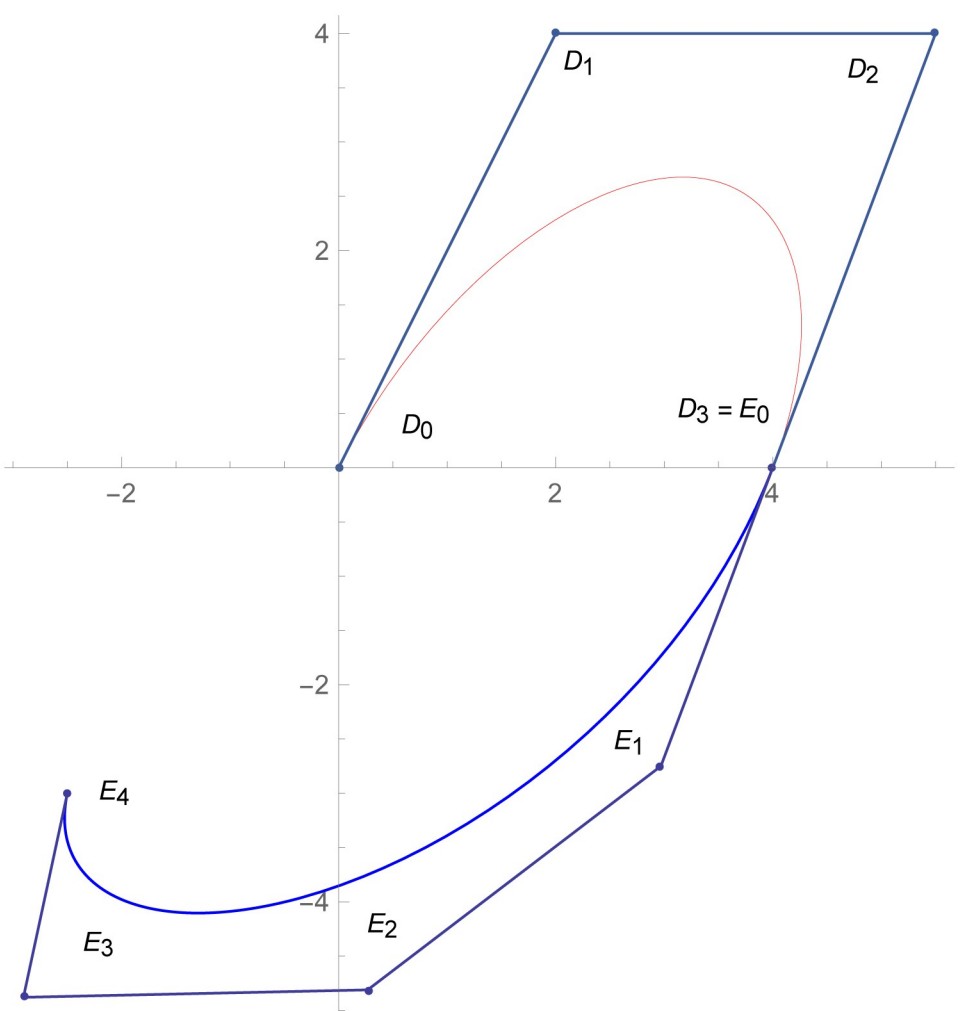

**Fig 9. Cubic-quantic Bézier-like curve segments connected by parametric $C^3$ continuity.**

Since $||\varphi s(\theta)|| = |\phi|||s(\theta)||$, we obtain:

$$\varphi = \psi\phi^3||W_2'(0)||^2(W_2'(0) \times W_2'''(0)) - 3\phi^6(W_2'(0) \times W_2''(0))(W_2'(0) \cdot W_2''(0)) \quad (21)$$

and by using the value of $\varphi$ in Eq (20), we have:

$$\kappa_1'(1) = \phi\left\{\frac{||W_2'(0)||^2(W_2'(0) \times W_2'''(0)) - 3(W_2'(0) \times W_2''(0))(W_2'(0) \cdot W_2''(0))}{||W_2'(0)||^5}\right\} = \phi\kappa_2'(0). \quad (22)$$

**Remark**: The curvature plot are useful to check the smooth continuity between the two curves which are connected by parametric and geometric continuity of any order. The curvature plot of the second curve $k_2(\tilde{w} - 1)$ can be drawn on $\tilde{w} \in [1, 2]$, the first curve's curvature $k_1(\tilde{w})$ can be plotted on $\tilde{w} \in [0, 1]$. The purpose of doing is just to clear visualization of curvature continuity.

**Example 2.1** Fig 7 *depicts the two quantic Bézier-like curves* $W(\tilde{w})$ *and* $W1(\tilde{w})$ *are connected by* $C^3$ *parametric continuity. The control points of the curve* $W(\tilde{w})$ *such that* $D_r$, *r = 0, 1, 2, 3, 4 are given by the choice of user and with the help of parametric* $C^3$ *constraints, given in*

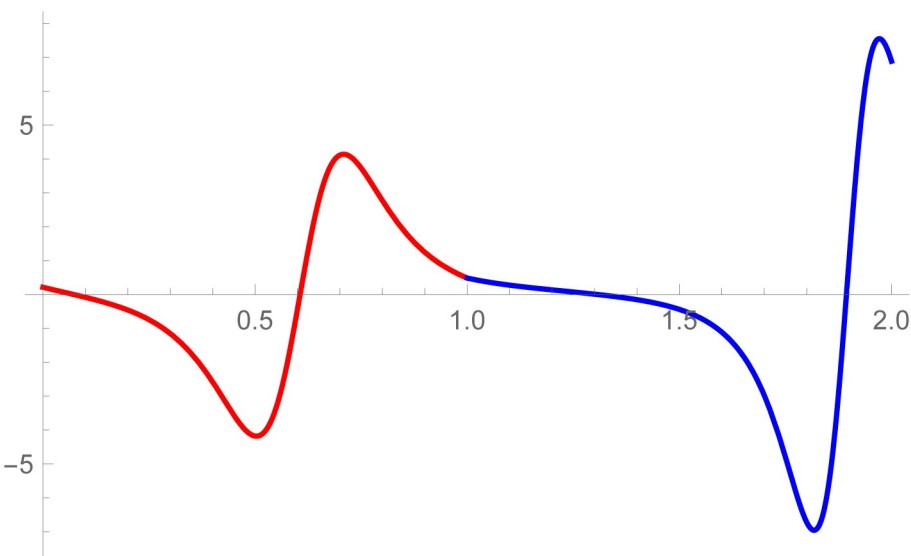

**Fig 10. Curvature plot of cubic-quantic Bézier-like curve segments.**

*the* Eqs (13) *and* (14), *are used to calculate the control points of second curve* $E_s$, *s* = 0, 1, 2, 3, 4. *At the end, the user provide the last control point of the second curve* $E_4$. *The curvature plot of that curves are expressed in the* Fig 8, *which shows the smooth continuity connection of these curves that are connected with parametric continuity.*

**Example 2.2** Fig 9 *shows the two different degrees of Bézier-like curves such that the curve* $W(\tilde{w})$ *is cubic and* $W1(\tilde{w})$ *is quantic Bézier-like curve. The control points of the curve* $W(\tilde{w})$ *such that* $D_r$, *r* = 0, 1, 2, 3 *are given by the choice of user and with the help of parametric* $C^3$ *constraints, given in the* Eqs (13) *and* (14), *are used to calculate the control points of second curve* $E_s$, *s* = 0, 1, 2, 3, 4. *At the end, the user provide the last control point of the second curve* $E_4$. *The curvature plot of that curves are expressed in the* Fig 10, *which shows the smooth continuity connection of these curves that are connected with parametric continuity.*

**Example 2.3** Fig 11 *shows that the two quantic Bézier-like curves* $W(\tilde{w})$ *and* $W1(\tilde{w})$ *are connected with* $G^3$ *geometric continuity. The control points of the curve* $W(\tilde{w})$ *such that* $D_r$, *r* = 0, 1, 2, 3, 4 *are given by the choice of user and with the help of geometric* $G^3$ *continuity constraints, given in the* Eqs (15) *and* (16), *are used to calculate the control points of second curve* $E_s$, *s* = 0, 1, 2, 3, 4. *At the end, the user provide the last control point of the second curve* $E_4$. *The curvature plot of that curves are expressed in the* Fig 12, *which shows the smooth continuity connection of these curves that are connected with geometric continuity.*

**Example 2.4** Fig 13 *demonstrates the two different degrees Bézier-like curves such that the cubic curve* $W(\tilde{w})$ *and* $W1(\tilde{w})$ *is quantic Bézier-like curve are connected with* $G^3$ *geometric continuity. The control points of the curve* $W(\tilde{w})$ *such that* $D_r$, *r* = 0, 1, 2, 3 *are given by the choice of user and with the help of geometric* $G^3$ *continuity constraints, given in the* Eqs (15) *and* (16), *are used to calculate the control points of second curve* $E_s$, *s* = 0, 1, 2, 3, 4. *At the end, the user provide the last control point of the second curve* $E_4$. *The curvature plot of that curves are expressed in the* Fig 14, *which shows the smooth continuity connection of these curves that are connected with geometric continuity.*

The above numerical examples show that the continuity constraints are valid and they can be used to connect any two different degree curves. By changing the value of shape parameters,

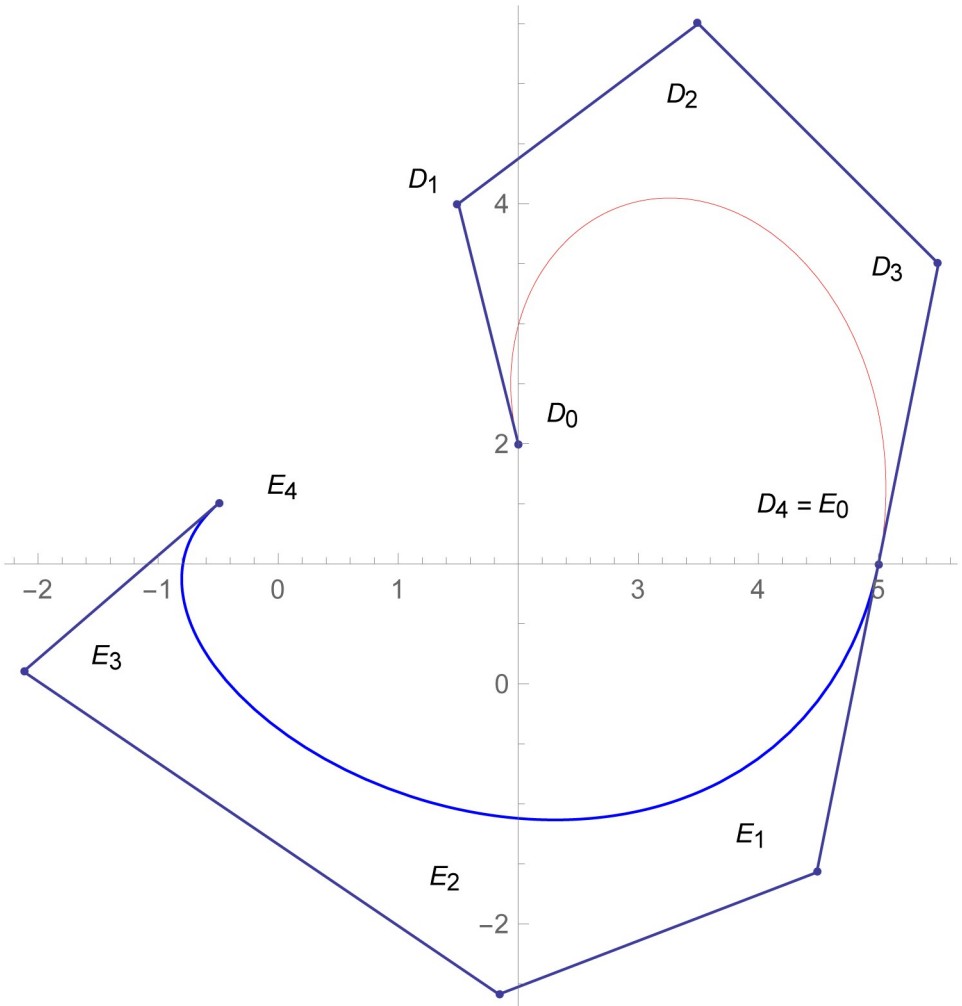

**Fig 11. Cubic-quantic Bézier-like curve segments with geometric $G^3$ continuity.**

curve can be altered by our choice. The curvature plot of that curve shows that the connection is smooth. The value of shape parameter effects the shape of the curve. Figs 15 and 16 show that after changing the value of shape parameter, the curve is also changed. It means that with the help of shape parameters, we can modify shape of the curve by our choice which is the prime property of the proposed gBC that is not present in the traditional Bézier models.

It is noted that, when the curve is connected by $C^3$ continuity, the first four control points of the second curve are calculated by the continuity constraints of $C^3$ and the remaining control points are given by the user but in the case of $C^2$ continuity just first three control points are calculated by continuity constraints and rest of all control points of the second curve are given by the user. Fig 17 depicts that the two Bézier-like curves are connected with $C^2$ continuity [26]. Similarly, Fig 18 shows that the two Bézier-like curves are connected with $G^2$ continuity [26]. To check the validation of the $C^2$ and $G^2$ continuity constraints, the curvature plot is illustrated. Figs 19 and 20 depict the curvature plot of the the parametric continuity $C^2$ and geometric continuity $G^2$ respectively, show the smooth connection of their joints.

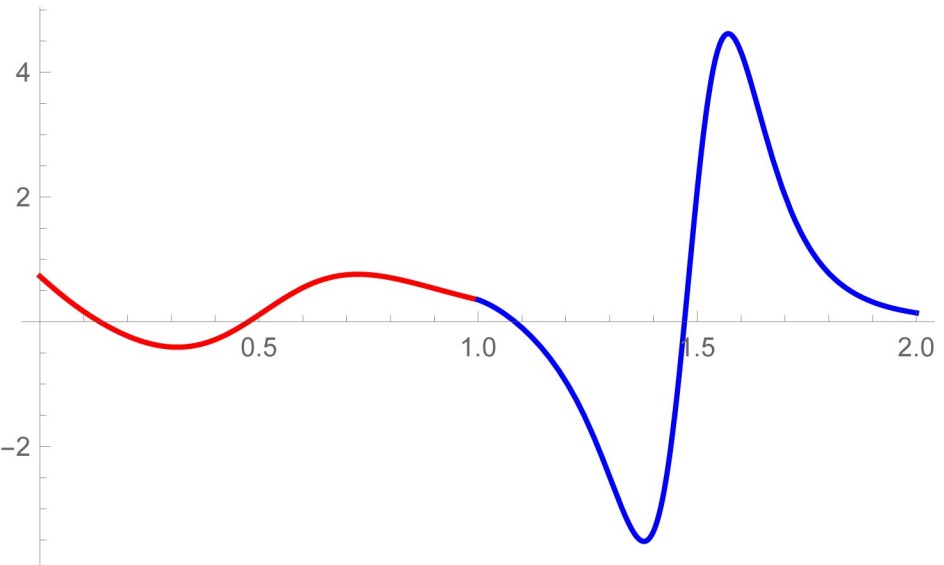

**Fig 12. Curvature plot of cubic-quantic Bézier-like curve segments.**

The $G^3$ continuity is smoother than $G^2$ continuity because $G^3$ continuity ensures that not only curvature of the curves aligned but their rate of change of curvature also matches. It means transition between the curves is not only visually smooth but also mathematically satisfied. In the setting of $G^2$ continuity, it is possible to notice visual imperfections or faults where the curves meet. In contrast, $G^3$ continuity reduces artifacts by providing smoother transitions. The curvature plot of the second curve $k_2(\tilde{w})$ is drawn on $\tilde{w} \in [1, 2]$ and the first curve's curvature $k_1(\tilde{w})$ is plotted on $\tilde{w} \in [1, 2]$. The purpose of doing this is just to clear visualization of curvature continuity. To verify that $G^3$ is smoother than $G^2$, curvature plot of the curves which is connected by $G^3$ and $G^2$ continuity are given in the Figs 12 and 20, respectively. It is clearly see in the Fig 12 that the curvature plot of the curves which is connected by $G^3$ is smoother than the curves which is connected by $G^2$ continuity in the Fig 20. Similarly, for the comparison of $C^3$ and $C^2$ continuity, the curvature plot of $C^3$ and $C^2$ are expressed in the Figs 8 and 19, respectively. It is clearly shown in the Figs 8 and 19, that the curvature plot which is connected by $C^3$ continuity are smoother than the curvature plot which are connected by $C^2$ continuity. Hence $G^3/C^3$ continuity are smoother than $G^2/C^2$ continuity.

## 3 Surface continuity

In this section, we will discuss the continuity of two surfaces.

### 3.1 $G^2$ Surface continuity

Ensuring the preservation of $G^2$ continuity between adjacent surface patches is of utmost importance in engineering applications. This implies that when two curved surfaces intersect, it is necessary for them to have a tangent plane in common or a normal line that is shared collectively at every point of intersection. There is a presence of continuity in four distinct

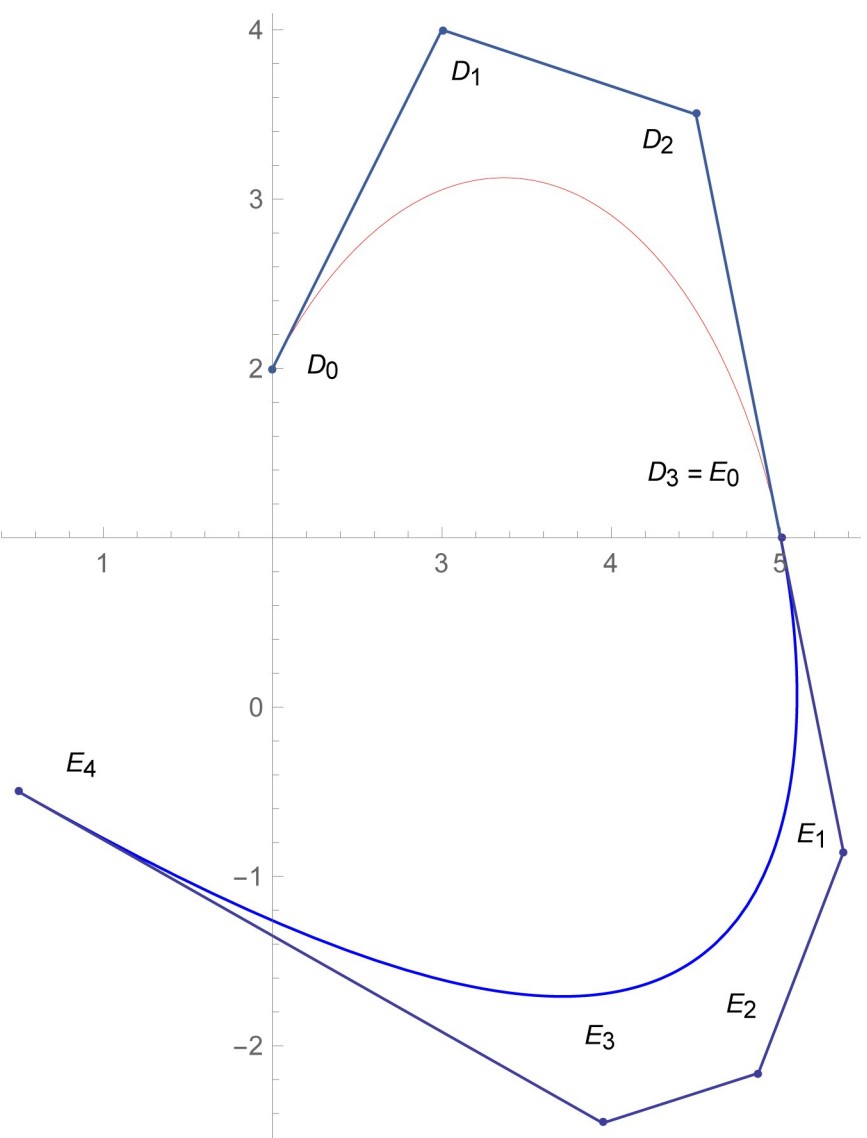

**Fig 13. Quantic geometric $G^3$ continuity.**

directions across two gBS $W_{r1,s1}(\tilde{w}, \tilde{w1}; \varsigma, \varsigma1, \zeta, \zeta1) = \sum\limits_{f=0}^{r1} \sum\limits_{f1=0}^{s1} D_{f,f1} \breve{u}_{f,r1}(\tilde{w}) \breve{u}_{f1,s1}(\tilde{w1})$ and

$W1_{r2,s2}(\tilde{w}, \tilde{w1}; \varsigma^*, \varsigma1^*, \zeta^*, \zeta1^*) = \sum\limits_{g=0}^{r2} \sum\limits_{g1=0}^{s2} D1_{g,g_1} \breve{u}_{g,r2}(\tilde{w}) \breve{u}_{g1,s2}(\tilde{w})$.

## 3.2 $G^2$ continuity in $\tilde{w}$ direction on every surface

**Theorem 3** *Consider two gBS, $W_{r1,s1}(\tilde{w}, \tilde{w1}; \varsigma, \varsigma1, \zeta, \zeta1)$ and $W1_{r2,s2}(\tilde{w}, \tilde{w1}; \varsigma^*, \varsigma1^*, \zeta^*, \zeta1^*)$, they meet the $G^2$ smooth continuity conditions in $\tilde{w}$ direction if they satisfy all the following*

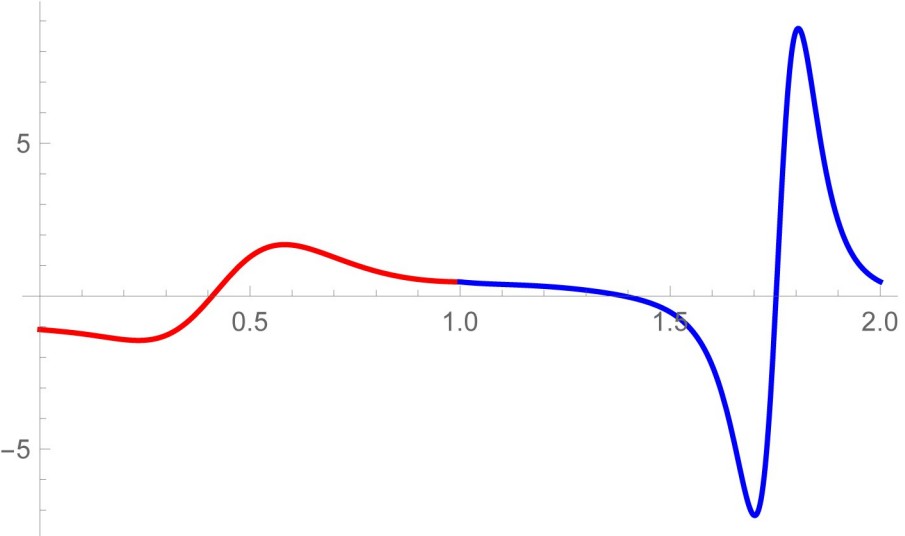

**Fig 14. Curvature plot of quantic geometric $G^3$ continuity.**

*conditions:*

$$
\begin{cases}
r_1 = r_2, \varsigma = \varsigma^*, \zeta = \zeta^*, \\
D_{f,r} = D1_{f,0}, \\
(s1 - 2 + \zeta1)(D_{f,s1} - D_{f,m1-1}) = z \cdot (s2 - 2 + \varsigma1^*)(D1_{f,1} - D1_{f,0}), \\
s1(2\zeta1 + s1 - 5)(D_{f,s1} - 2D_{f,s1-1} + D_{f,s1-2}) + 2(-6 + 2\zeta1 + \varsigma1)(D_{f,s1-1} - D_{f,s1-2}) \\
= z^2[s2(2\varsigma1^* + s2 - 5)(D_{f,0} - 2D_{f,1} + D_{f,2}) + 2(-6 + 2\varsigma1^* + \zeta1^*)(D_{f,1} - D_{f,2})], \\
(f = 0, 1, 2, \cdots, r_1),
\end{cases}
\tag{23}
$$

*where $z > 0$ is any positive real value.*

   **Proof** To achieve $G^0$ continuity, it is required for both areas of gBS to possess a shared edge curve i.e.,

$$
W_{r1,s1}(\tilde{w}, 1; \varsigma, \varsigma1, \zeta, \zeta1) = W1_{r2,s2}(\tilde{w}, 0; \varsigma^*, \varsigma1^*, \zeta^*, \zeta1^*),
\tag{24}
$$

which results

$$
\sum_{f=0}^{r1} D_{f,s1} \breve{u}_{f,r1}(\tilde{w}, \varsigma, \zeta) = \sum_{g=0}^{r2} D1_{g,0} \breve{u}_{g,r2}(\tilde{w}, \varsigma^*, \zeta^*),
\tag{25}
$$

implying

$$
\sum_{f=0}^{r} D_{f,r} \breve{u}_{f,r}(\tilde{w}, \varsigma, \zeta) = \sum_{f=0}^{r} D1_{f,0} \breve{u}_{f,r}(\tilde{w}, \varsigma^*, \zeta^*).
\tag{26}
$$

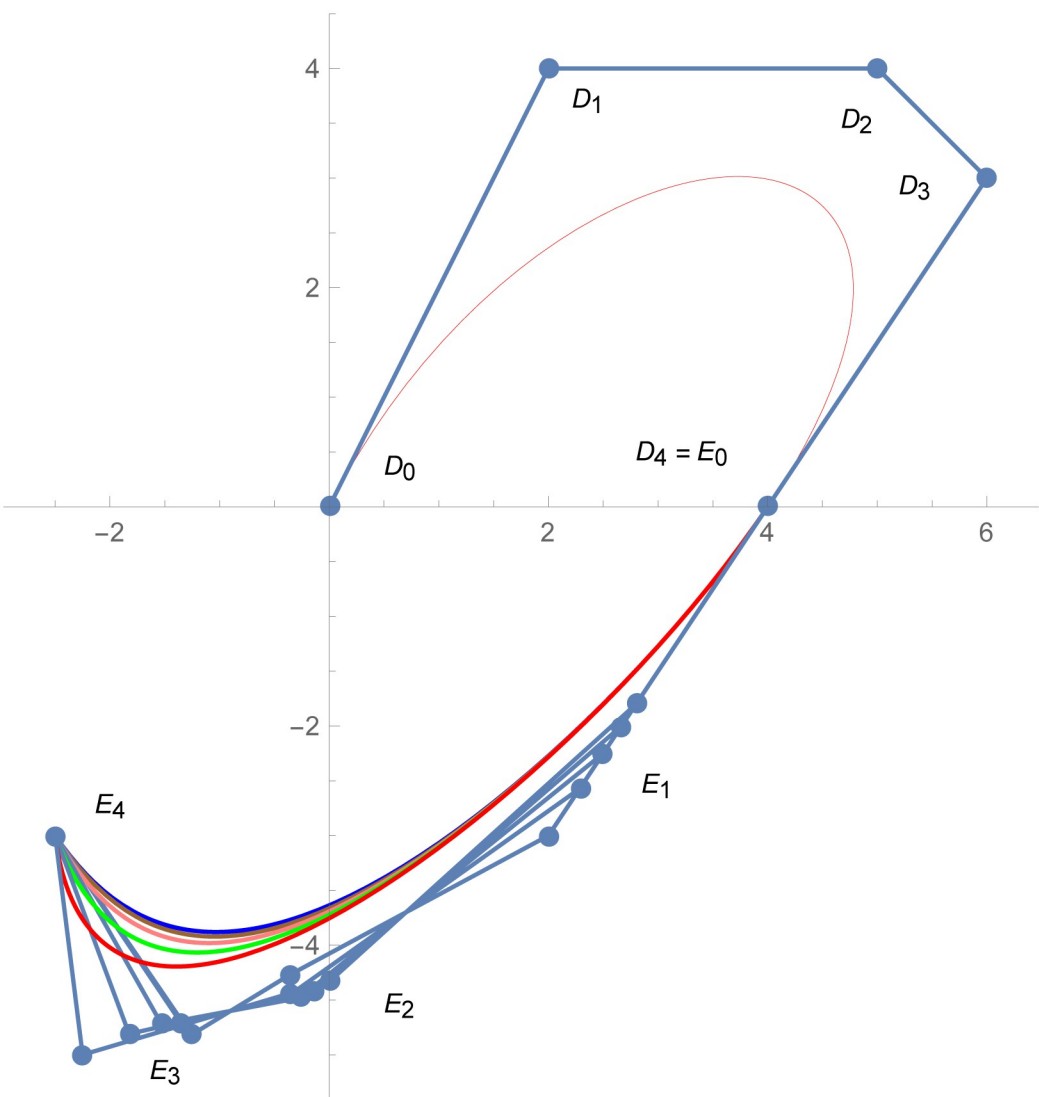

**Fig 15. Effect on quantic parametric $C^3$ continuity curves by changing the value of shape parameters.**

If $\varsigma = \varsigma^*$ and $\zeta = \zeta^*$, then

$$D_{f,r} = D1_{f,0}. \tag{27}$$

The normal direction fails to split off at the boundary for either of the two parts of the surface; hence, the connecting boundary must have a common tangent plane in order to preserve $G^1$ continuity between both parts of the surface. So, we have the following condition:

$$\frac{\partial}{\partial \tilde{w}} W_{r,s1}(\tilde{w}, 1; \varsigma, \varsigma 1, \zeta, \zeta 1) \times \frac{\partial}{\partial \tilde{w1}} W_{r,s1}(\tilde{w}, 1; \varsigma, \varsigma 1, \zeta, \zeta 1) =$$

$$z(\tilde{w1}) \frac{\partial}{\partial \tilde{w}} W1_{r,s2}(\tilde{w}, 1; \varsigma^*, \varsigma 1^*, \zeta^*, \zeta 1^*) \times \frac{\partial}{\partial \tilde{w1}} W1_{r,s2}(\tilde{w}, 1; \varsigma^*, \varsigma 1^*, \zeta^*, \zeta 1^*),$$

where $z(\tilde{w1})$ is the scaling factor among their normal vectors such that $z(\tilde{w1}) > 0$.

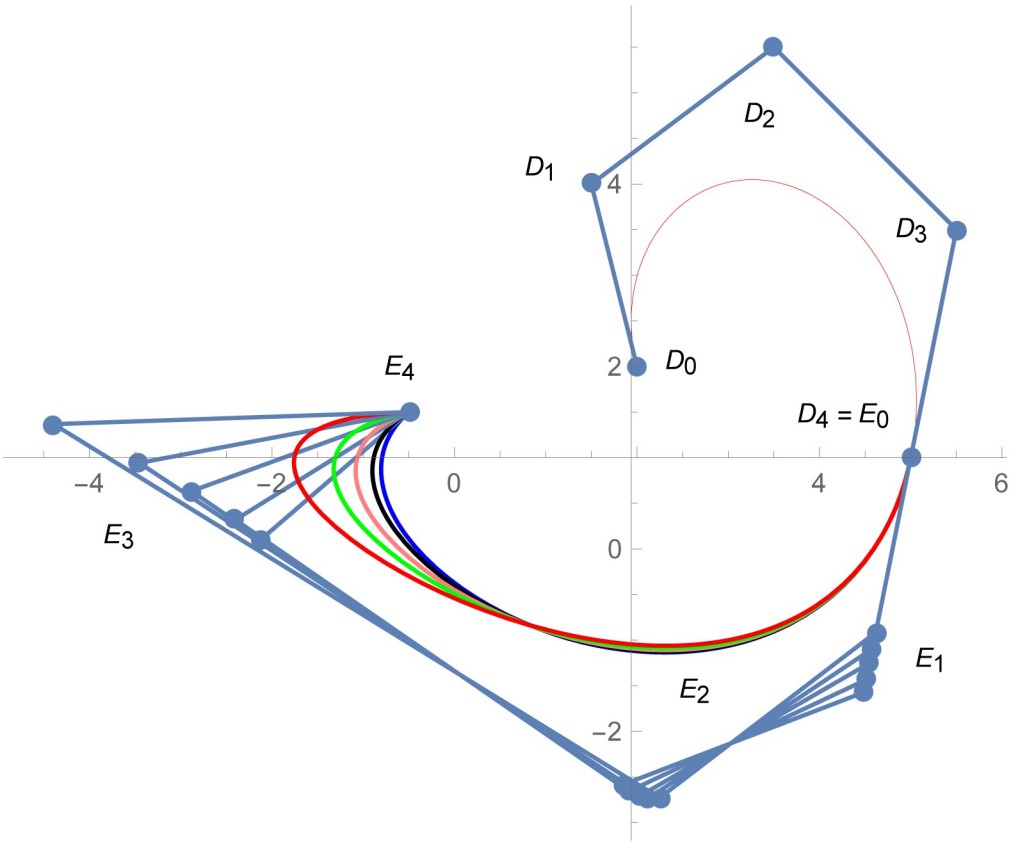

**Fig 16. Effect on cubic-quantic geometric $G^3$ continuity curves by changing the value of shape parameters.**

Equivalently,

$$\frac{\partial}{\partial \tilde{w}} W_{r,s1}(\tilde{w}, 1; \varsigma, \varsigma 1, \zeta, \zeta 1) = z \cdot \frac{\partial}{\partial \tilde{w1}} W1_{r,s2}(\tilde{w}, 0; \varsigma^*, \varsigma 1^*, \zeta^*, \zeta 1^*). \tag{28}$$

Since

$$\frac{\partial}{\partial \tilde{w}} W_{r,s1}(\tilde{w}, 1; \varsigma, \varsigma 1, \zeta, \zeta 1) = \sum_{f=0}^{r} \breve{u}_{f,r}(\tilde{w}, \varsigma, \zeta)(s1 - 2 + \zeta 1)(D_{f,s1} - D_{f,s1-1}) \tag{29}$$

and

$$\frac{\partial}{\partial \tilde{w}} W1_{r,s2}(\tilde{w}, 0; \varsigma^*, \varsigma 1^*, \zeta^*, \zeta 1^*) = \sum_{f=0}^{r} \breve{u}_{f,n}(\tilde{w}, \varsigma^*, \zeta^*)(s2 - 2 + \varsigma 1^*)(D1_{f,1} - D1_{f,0}). \tag{30}$$

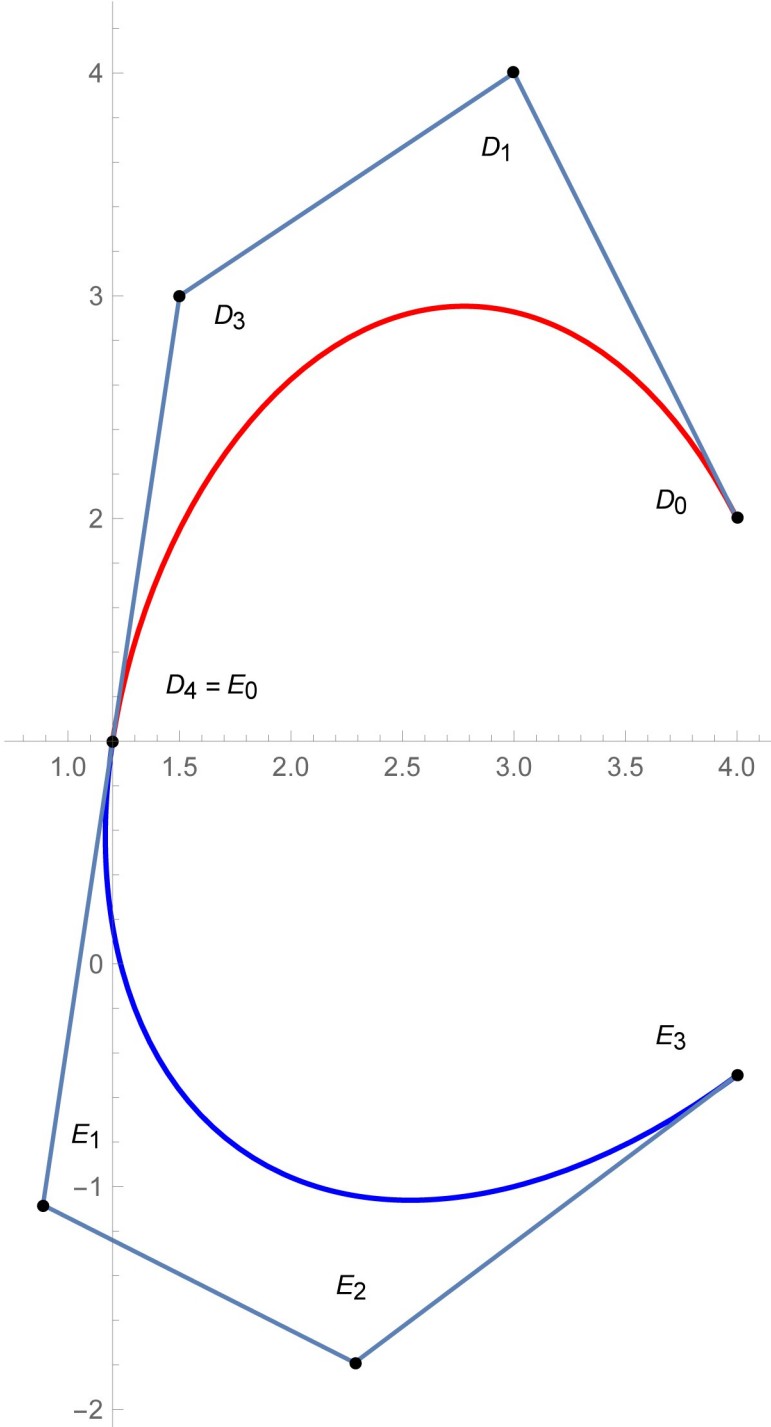

**Fig 17. Two Bézier-like curves connected by $C^2$ parametric continuity.**

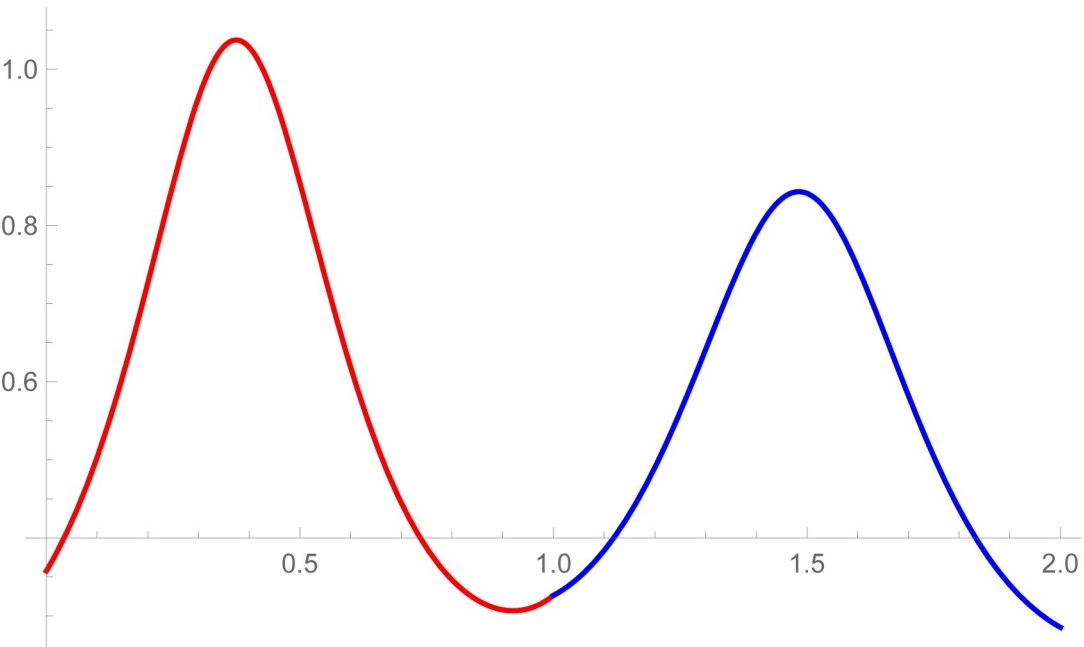

**Fig 18. Two Bézier-like curves connected by $G^2$ geometric continuity.**

Substituting Eqs (29) and (30) in Eq (28), we have

$$\sum_{f=0}^{r} \breve{u}_{f,r}(\tilde{w}, \varsigma, \zeta)(s1 - 2 + \zeta 1)(D_{f,s1} - D_{f,s1-1}) = z \cdot \sum_{f=0}^{r} \breve{u}_{f,r}(\tilde{w}, \varsigma^*, \zeta^*)(s2 \tag{31}$$
$$-2 + \varsigma 1^*)(D1_{f,1} - D1_{f,0}).$$

If $\varsigma = \varsigma^*$ and $\zeta = \zeta^*$, then

$$(s1 - 2 + \zeta 1)(D_{f,s1} - D_{f,m1-1}) = z \cdot (s2 - 2 + \varsigma 1^*)(D1_{f,1} - D1_{f,0}). \tag{32}$$

By using Eq (27) in Eq (32) and after some simplifications, we have

$$D1_{i,1} = D_{f,r} + \frac{(s1 - 2 + \zeta 1)}{(s2 - 2 + \varsigma 1^*)}(D_{f,s1} - D_{f,s1-1}). \tag{33}$$

Eqs (27) and (32), explain the continuity conditions on each surface, are the general conditions for $G^1$ continuity in the $\tilde{w}$ direction. In accumulation, the two surfaces must maintain the same normal curvature at any point on their joint boundary in order to satisfy the $G^1$ continuity condition. As a result, the two surfaces must also satisfy the $G^2$ continuity constraints:

$$\frac{\partial^2}{\partial \tilde{w}1^2} W_{r,s1}(\tilde{w}, 1; \varsigma, \varsigma 1, \zeta, \zeta 1) = z^2 \frac{\partial^2}{\partial \tilde{w}^2} W1_{r,s2}(\tilde{w}, 0; \varsigma^*, \varsigma 1^*, \zeta^*, \zeta 1^*) + 2zk(\tilde{w})$$

$$\frac{\partial^2}{\partial \tilde{w} \partial \tilde{w}1} W1_{r,s2}(\tilde{w}, 0; \varsigma^*, \varsigma 1^*, \zeta^*, \zeta 1^*) + k^2(\tilde{w}) \frac{\partial^2}{\partial \tilde{w}1 \partial \tilde{w}1} W1_{r,s2}(\tilde{w}, 0; \varsigma^*, \varsigma 1^*, \zeta^*, \zeta 1^*)$$

$$+ c \frac{\partial}{\partial \tilde{w}} W1_{r,s2}(\tilde{w}, 0; \varsigma^*, \varsigma 1^*, \zeta^*, \zeta 1^*) + d(\tilde{w}) \frac{\partial}{\partial \tilde{w}} W1_{r,s2}(\tilde{w}, 0; \varsigma^*, \varsigma 1^*, \zeta^*, \zeta 1^*),$$

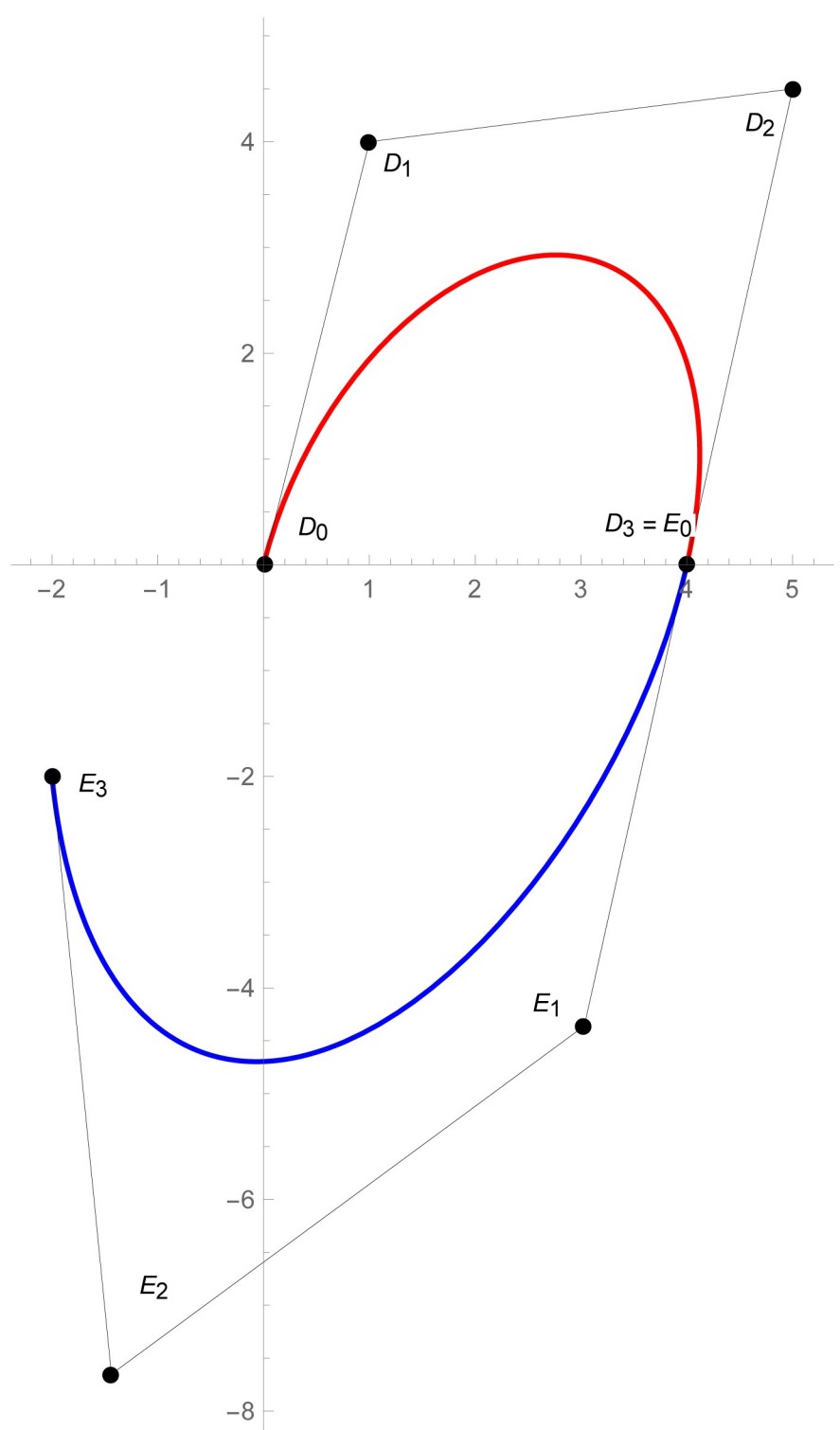

**Fig 19. Curvature plot of $C^2$ continuity curve segments.**

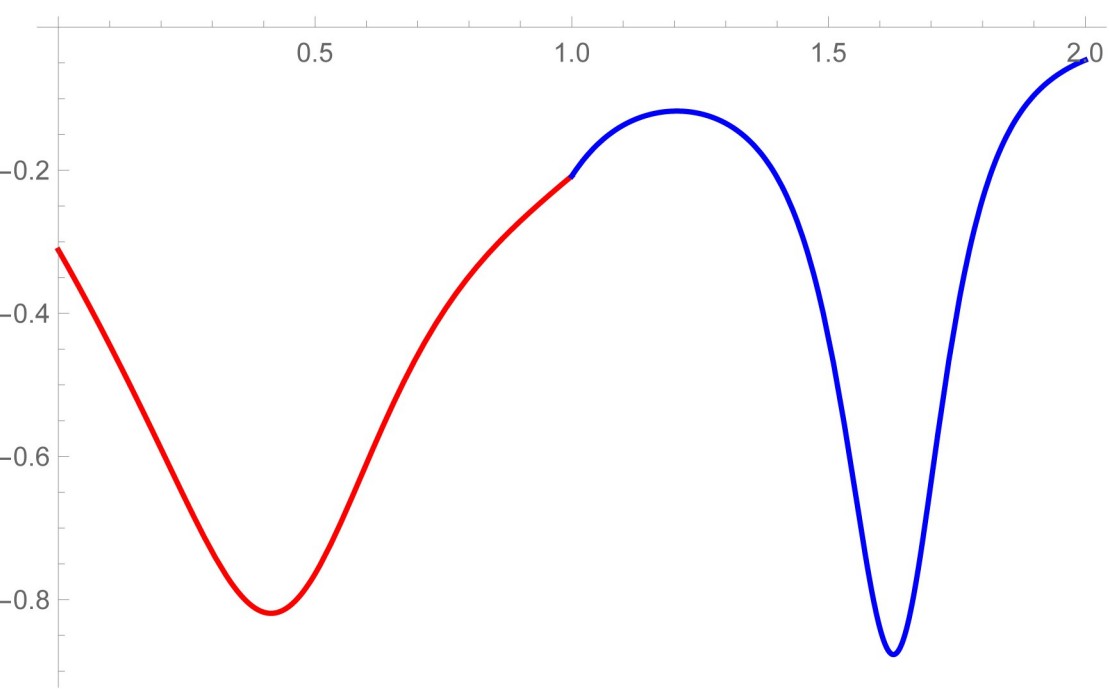

**Fig 20. Curvature plot of $G^2$ continuity curve segments.**

where the functions $k(\tilde{w})$, $d(\tilde{w})$ can be expressed as linear functions of $\tilde{w}$ and $c$ represents an indeterminate constant. In order to simplify the equation and facilitate calculations in the context of the applied submission, it is common practice to assume that both $k(\tilde{w})$ and $d(\tilde{w})$ are equal to zero. Consequently, the aforementioned equation can be further reduced as:

$$\frac{\partial^2}{\partial \tilde{w1}^2} W_{r,s1}(\tilde{w}, 1; \varsigma, \varsigma1, \zeta, \zeta1) = z^2 \frac{\partial^2}{\partial \tilde{w1}^2} W1_{r,s2}(\tilde{w}, 0; \varsigma^*, \varsigma1^*, \zeta^*, \zeta1^*). \qquad (34)$$

Since

$$\frac{\partial^2}{\partial \tilde{w1}^2} W_{r,s1}(\tilde{w}, 1; \varsigma, \varsigma1, \zeta, \zeta1) = \sum_{f=0}^{n} \breve{u}_{f,n}(\tilde{w}, \varsigma, \zeta)[s1(2\zeta1 + m1 - 5)(D_{f,s1} - 2D_{f,s1-1}$$
$$+ D_{f,s1-2}) + 2(-6 + 2\zeta1 + \varsigma1)(D_{f,s1-1} - D_{f,s1-2})] \qquad (35)$$

and

$$\frac{\partial^2}{\partial \tilde{w1}^2} W1_{r,s2}(\tilde{w}, 0; \varsigma^*, \varsigma1^*, \zeta^*, \zeta1^*) = \sum_{f=0}^{r} \breve{u}_{f,r}(\tilde{w}, \varsigma^*, \zeta^*)[s2(2\varsigma1^* + s2 - 5)(D_{f,0}$$
$$- 2D_{f,1} + D_{f,2}) + 2(-6 + 2\varsigma1^* + \zeta1^*)(D_{f,1} - D_{f,2})]. \qquad (36)$$

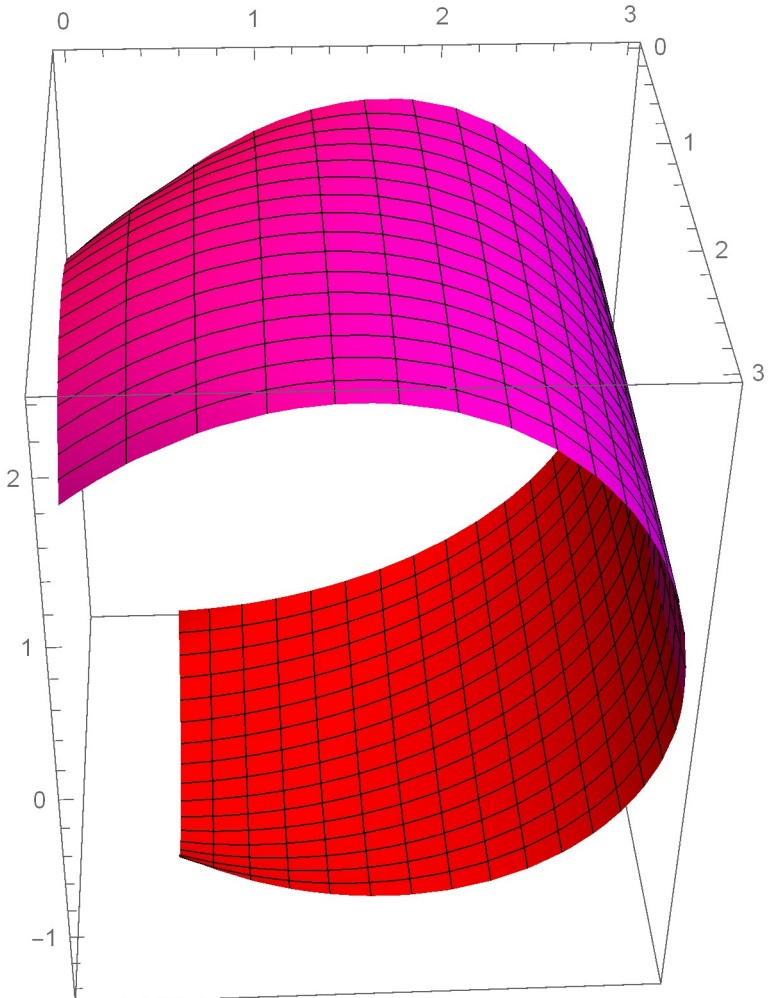

**Fig 21. The cubic Bézier- like surface patches connected by $G^2$ geometric surface continuity.**

Substituting Eqs (35) and (36) in Eq (34), we achieve

$$\sum_{f=0}^{r} \breve{u}_{f,n}(\tilde{w}, \varsigma, \zeta)[s1(2\zeta1 + s1 - 5)(D_{f,s1} - 2D_{f,s1-1} + D_{f,s1-2}) + 2(-6 + 2\zeta1$$
$$+\varsigma1)(D_{f,s1-1} - D_{f,s1-2})] = z^2[\sum_{f=0}^{r} \breve{u}_{f,r}(t, \varsigma^*, \zeta^*)[s2(2\varsigma1^* + s2 - 5)(D_{f,0}$$
$$-2D_{f,1}x + D_{f,2}) + 2(-6 + 2\varsigma1^* + \zeta1^*)(D_{f,1} - D_{f,2})]].$$

(37)

If $\varsigma = \varsigma^*$ and $\zeta = \zeta^*$, then

$$s1(2\zeta1 + s1 - 5)(D_{f,s1} - 2D_{f,s1-1} + D_{f,s1-2}) + 2(-6 + 2\zeta1 + \varsigma1)(D_{f,s1-1}$$
$$-D_{f,s1-2}) = z^2[s2(2\varsigma1^* + s2 - 5)(D_{f,0} - 2D_{f,1} + D_{f,2}) + 2(-6 + 2\varsigma1^*$$
$$+\zeta1^*)(D_{f,1} - D_{f,2})].$$

(38)

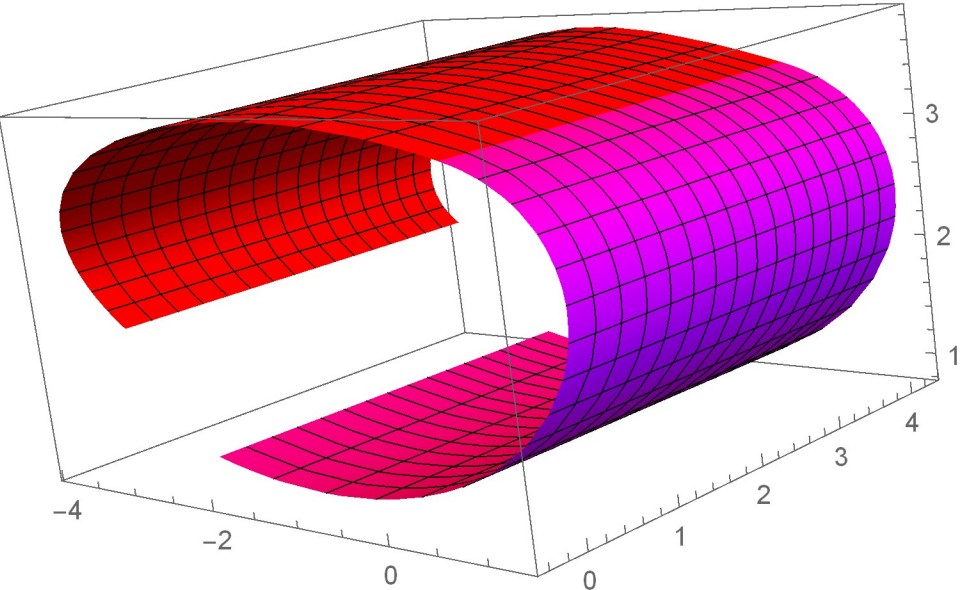

**Fig 22. The quantic Bézier- like surface patches joined by $G^2$ geometric surface continuity.**

To sum up, if the two surfaces satisfy Eqs (27) and (33) and (38) at the same time, the $G^2$ continuity in $\tilde{w}$ direction is achieved.

### 3.3 $G^2$ continuity in $\tilde{w}$ and $\tilde{w}1$ directions on every surface

**Theorem 4** *Consider two gBS, $W_{r1,s1}(\tilde{w}, \tilde{w}1; \varsigma, \varsigma1, \zeta, \zeta1)$ and $W1_{r2,s2}(\tilde{w}, \tilde{w}1; \varsigma^*, \varsigma1^*, \zeta^*, \zeta1^*)$, they meet the $G^2$ smooth continuity conditions in $\tilde{w}$ and $\tilde{w}1$ directions if they satisfy all the following conditions:*

$$
\begin{cases}
r_1 = s_2, \\
\varsigma = \varsigma1^*, \zeta = \zeta1^*, \\
D_{f,r} = D1_{0,g}, \\
(s1 - 2 + \zeta1)(D_{f,s1} - D_{f,s1-1}) = z \cdot (r2 - 2 + \varsigma^*)(D1_{1,g} - D1_{0,g}), \\
s1(2\zeta1 + s1 - 5)(D_{f,s1} - 2D_{f,s1-1} + D_{f,s1-2}) + 2(-6 + 2\zeta1 + \varsigma1)(D_{f,s1-1} - D_{f,s1-2}) \\
= z^2[s2(2\varsigma^* + s2 - 5)(D_{f,0} - 2D_{f,1} + D_{f,2}) + 2(-6 + 2\varsigma^* + \zeta^*)(D_{f,1} - D_{f,2})].
\end{cases}
\tag{39}
$$

The condition in which $W_{r1,s1}(\tilde{w}, \tilde{w}1; \varsigma, \varsigma1, \zeta, \zeta1)$ has continuity in $\tilde{w}$ direction and the surface $W1_{r2,s2}(\tilde{w}, \tilde{w}1; \varsigma^*, \varsigma1^*, \zeta^*, \zeta1^*)$ has continuity in $\tilde{w}_1$ direction, is

$$
W_{r1,s1}(\tilde{w}, 1; \varsigma, \varsigma1, \zeta, \zeta1) = W1_{r2,s2}(0, \tilde{w}1; \varsigma^*, \varsigma1^*, \zeta^*, \zeta1^*)
\tag{40}
$$

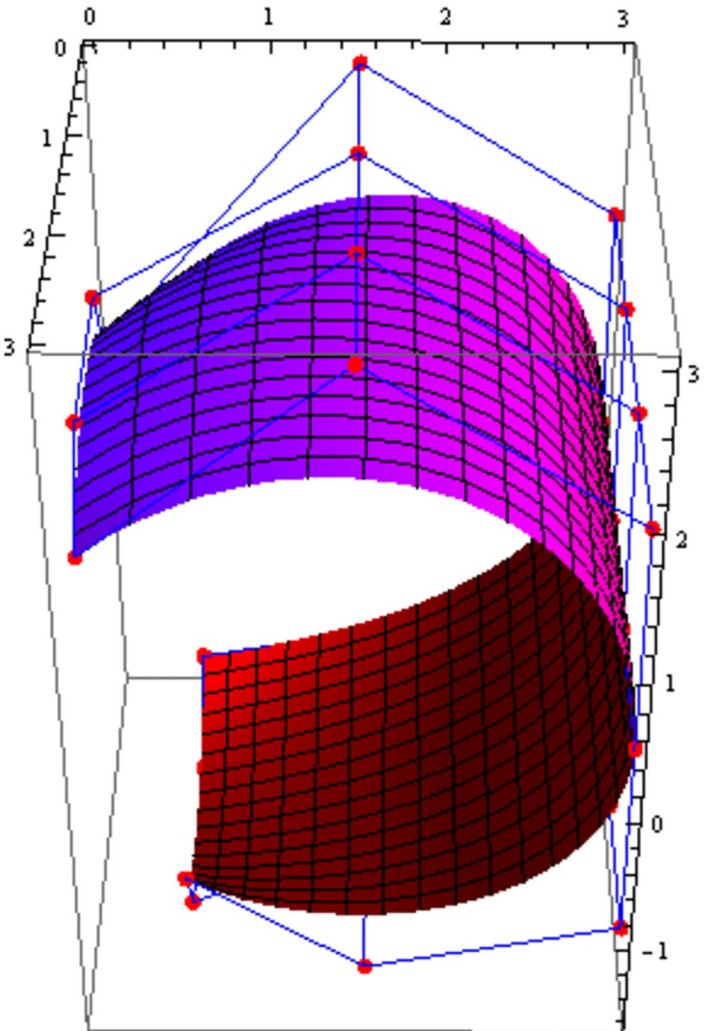

**Fig 23. The cubic Bézier- like surfaces for $f$ = 1.5, $\varsigma = \varsigma^*$ = 1.5, $\zeta = \zeta^*$ = 1.6.**

or

$$\sum_{f=0}^{r1} D_{f,s1} \breve{u}_{f,r1}(\tilde{w}, \varsigma, \zeta) = \sum_{g=0}^{r2} D1_{0,g} \breve{u}_{g,r2}(\tilde{w}1, \varsigma1^*, \zeta1^*), \tag{41}$$

implying

$$\sum_{f=0}^{r} D_{f,r} \breve{u}_{f,r}(\tilde{w}, \varsigma, \zeta) = \sum_{f=0}^{r} D1_{0_f} \breve{u}_{f,r}(\tilde{w}, \varsigma1^*, \zeta1^*). \tag{42}$$

If $\varsigma = \varsigma1^*$ and $\zeta = \zeta1^*$, then

$$D_{f,r} = D1_{0_f}. \tag{43}$$

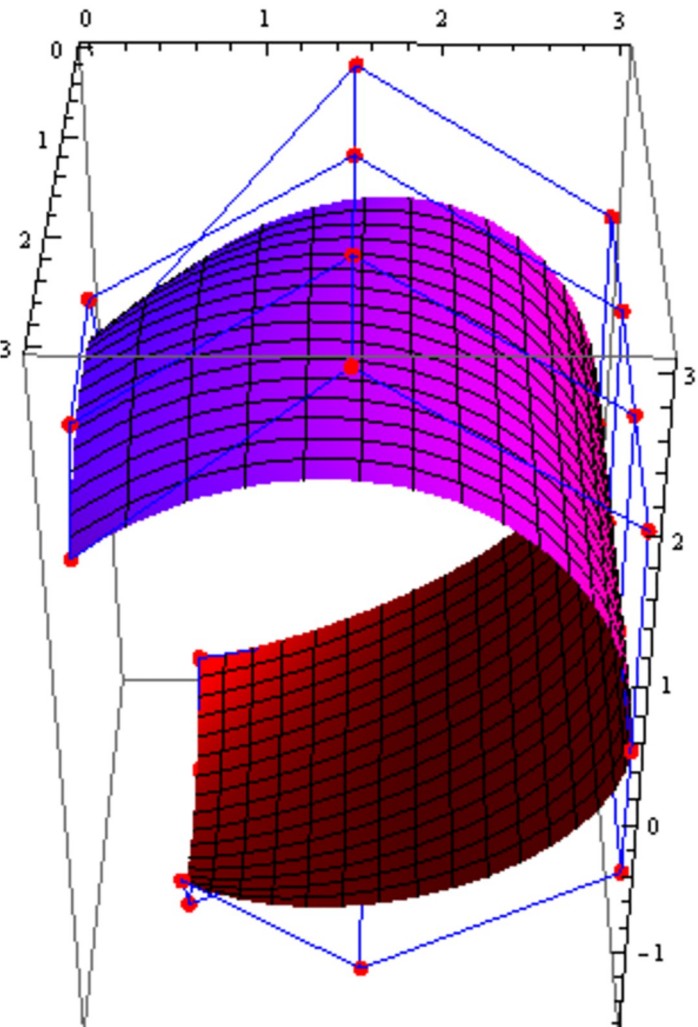

**Fig 24. The cubic Bézier- like surfaces for $f = 1.5$, $\varsigma = \varsigma^* = 1.8$, $\zeta = \zeta^* = 1.9$.**

In order to establish the existence of a common tangent plane, the following condition must be satisfied:

$$\frac{\partial}{\partial \widetilde{w1}} W_{r,s1}(\tilde{w}, 1; \varsigma, \varsigma1, \zeta, \zeta1) = z \cdot \frac{\partial}{\partial \tilde{w}} W1_{r,s2}(0, \tilde{w1}; \varsigma^*, \varsigma1^*, \zeta^*, \zeta1^*). \tag{44}$$

Since

$$\frac{\partial}{\partial t1} W_{r,s1}(\tilde{w}, 1; \varsigma, \varsigma1, \zeta, \zeta1) = \sum_{f=0}^{r} \breve{u}_{f,r}(\tilde{w}, \varsigma, \zeta)(s1 - 2 + \zeta1)(D_{f,s1} - D_{f,s1-1}) \tag{45}$$

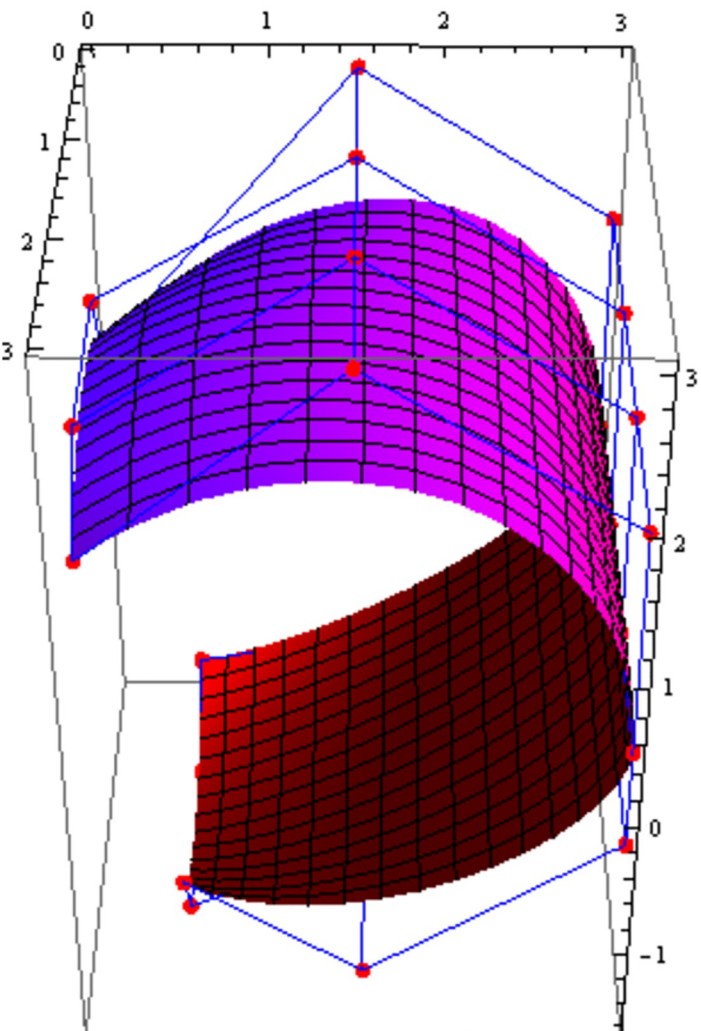

**Fig 25. The cubic Bézier- like surfaces for $f$ = 1.5, $\varsigma = \varsigma^* = 2.2$, $\zeta = \zeta^* = 2.3$.**

and

$$\frac{\partial}{\partial \tilde{w}} W1_{r2,s2}(0, \tilde{w1}; \varsigma^*, \varsigma1^*, \zeta^*, \zeta1^*) = \sum_{g1=0}^{s2} \breve{u}_{g,s2}(\tilde{w1}, \varsigma1^*, \zeta1^*)(n2 - 2 + \varsigma^*)(D1_{1,g} - D1_{0,g}). \quad (46)$$

Substituting Eqs (45) and (46) in Eq (44), we have

$$\sum_{f=0}^{r} \breve{u}_{f,r}(\tilde{w}, \varsigma, \zeta)(s1 - 2 + \zeta1)(D_{f,s1} - D_{f,s1-1}) = z \cdot \sum_{g1=0}^{s2} \breve{u}_{g,s2}(\tilde{w1}, \varsigma1^*, \zeta1^*)(n2 \\ -2 + \varsigma^*)(D1_{1,g} - D1_{0,g}). \quad (47)$$

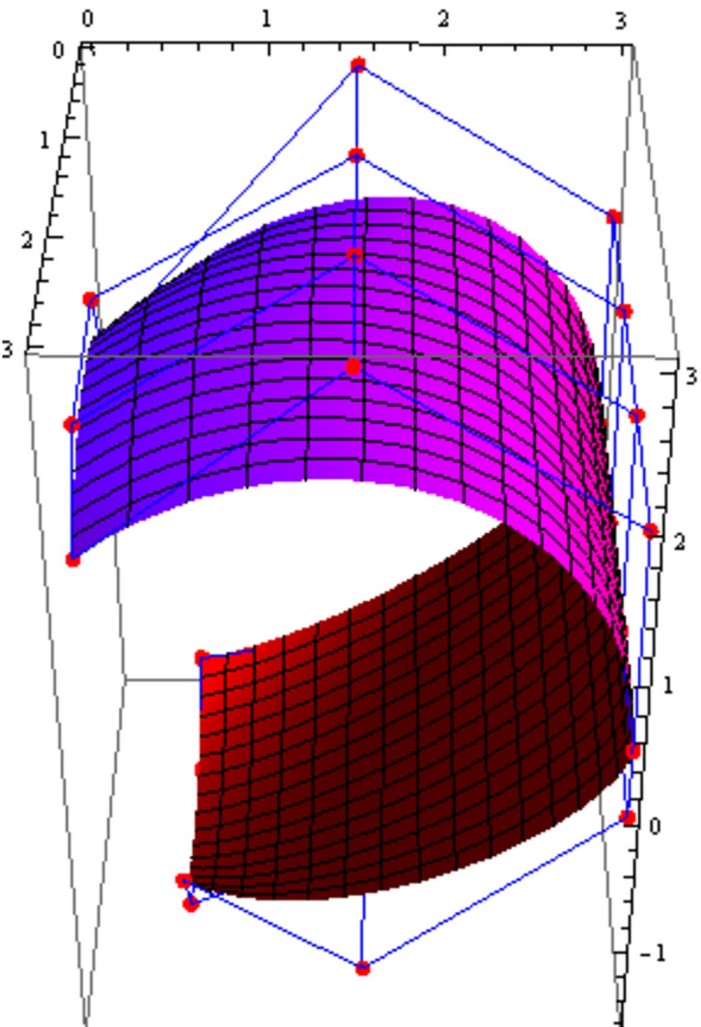

**Fig 26. The cubic Bézier- like surfaces for $f$ = 1.5, $\varsigma = \varsigma^*$ = 2.7, $\zeta = \zeta^*$ = 2.8.**

If $\varsigma = \varsigma 1^*$ and $\zeta = \zeta 1^*$, then

$$(s1 - 2 + \zeta 1)(D_{f,s1} - D_{f,s1-1}) = z \cdot (r2 - 2 + \varsigma^*)(D1_{1,g} - D1_{0,g}). \tag{48}$$

Therefore, the equations representing the general conditions for $G^1$ continuity in the $\tilde{w}$ and $\tilde{w}$ directions are given by the Eqs (38) and (44); when $\varsigma = \varsigma 1^*$ and $\zeta = \zeta 1^*$, the continuity requirements can be found in Eqs (43) and (48). In addition to the $G^1$ smooth continuity conditions, it is also necessary for the two surfaces to exhibit identical normal curvatures at every point along their common boundary. Therefore, in order to achieve $G^2$ continuity, they must fulfil this requirement:

$$\frac{\partial^2}{\partial \tilde{w1}^2} W_{r,s1}(\tilde{w}, 1; \varsigma, \varsigma 1, \zeta, \zeta 1) = z^2 \frac{\partial^2}{\partial \tilde{w}^2} W1_{r,s2}(0, \tilde{w1}; \varsigma^*, \varsigma 1^*, \zeta^*, \zeta 1^*). \tag{49}$$

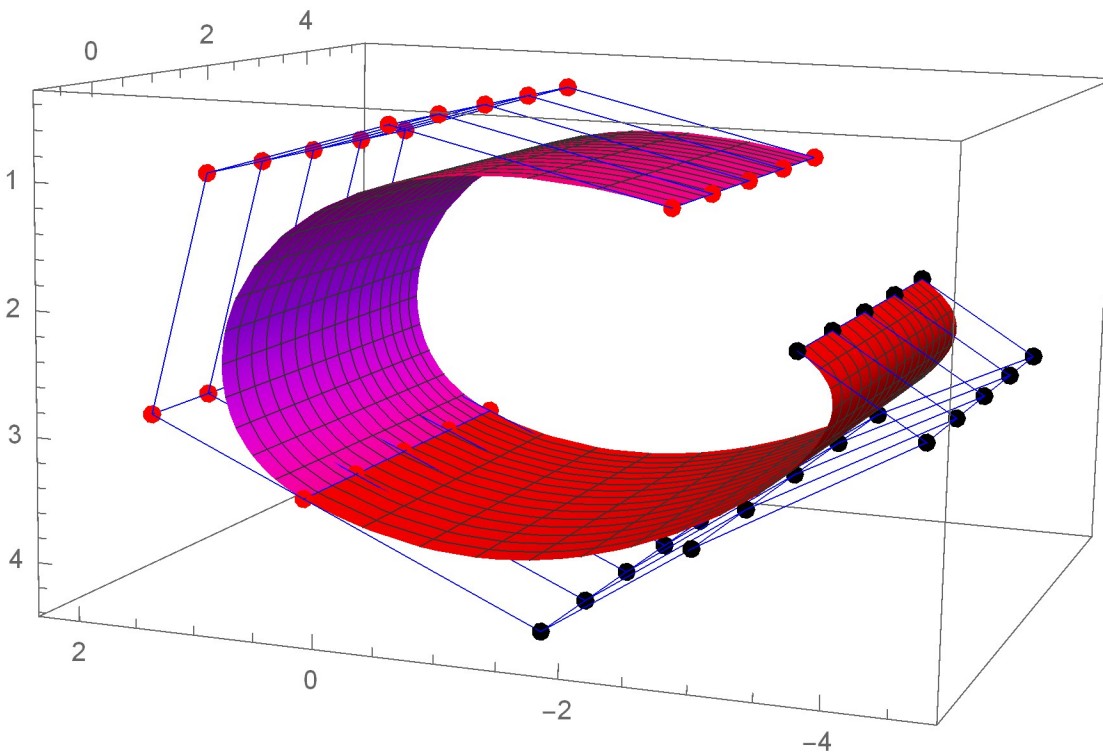

**Fig 27. The quantic Bézier-like surface for** $f$ = 0.9, $\varsigma = \varsigma^* = 1.2$, $\zeta = \zeta^* = 1.2$.

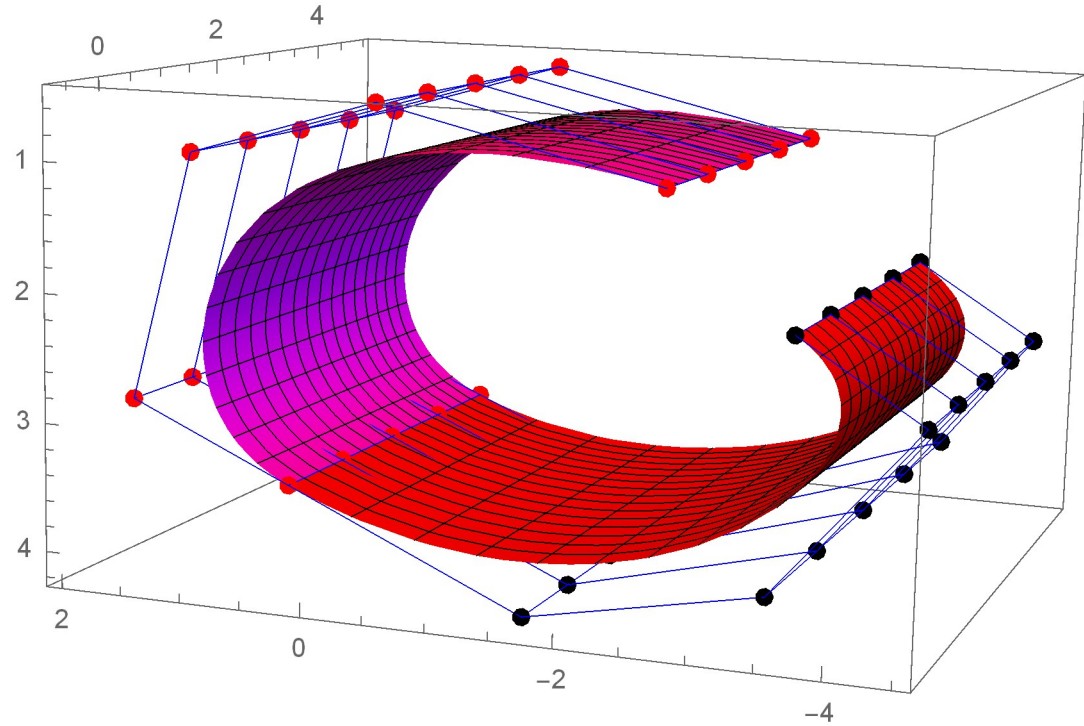

**Fig 28. The quantic Bézier- like surface for** $f$ = 0.9, $\varsigma = \varsigma^* = 1.5$, $\zeta = \zeta^* = 1.5$.

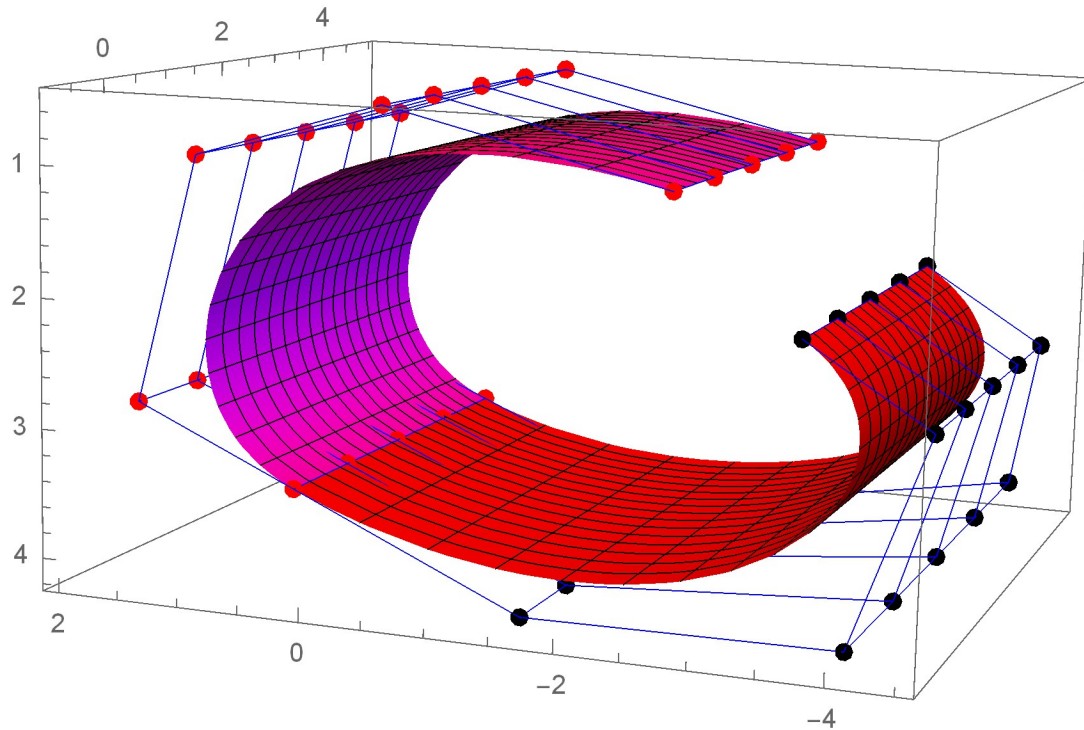

**Fig 29. The quantic Bézier- like surface for** $f = 0.9$, $\varsigma = \varsigma^* = 1.8$, $\zeta = \zeta^* = 1.8$.

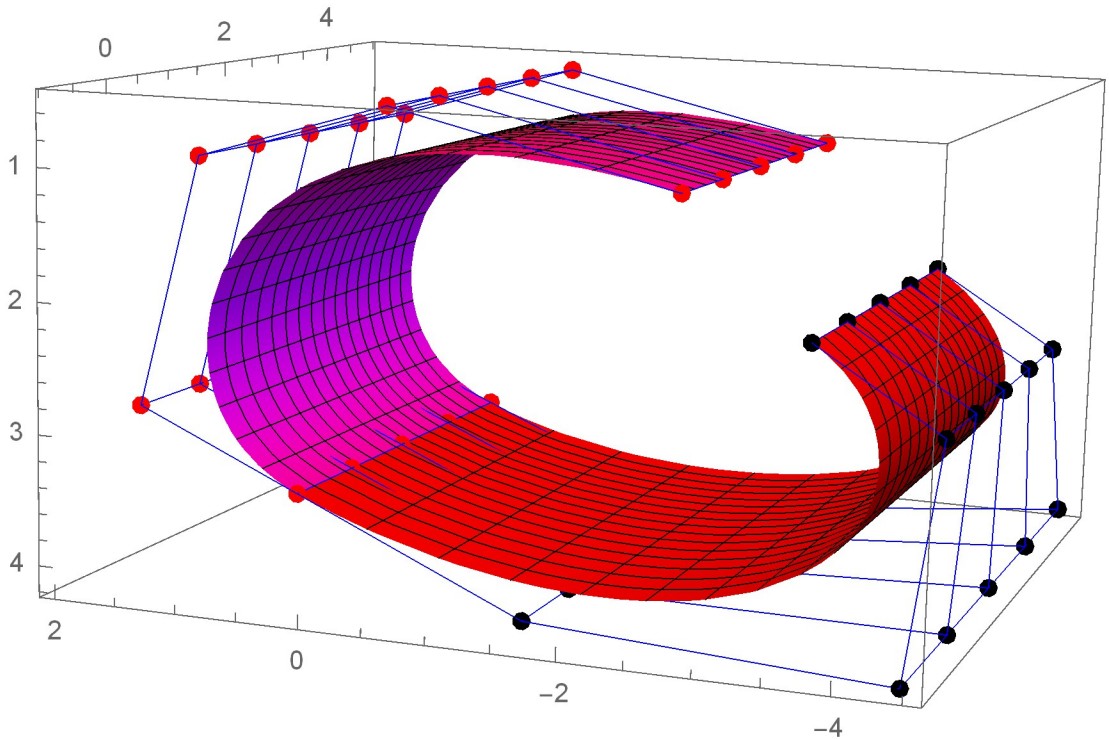

**Fig 30. The quantic Bézier- like surface for** $f = 0.9$, $\varsigma = \varsigma^* = 2$, $\zeta = \zeta^* = 2$.

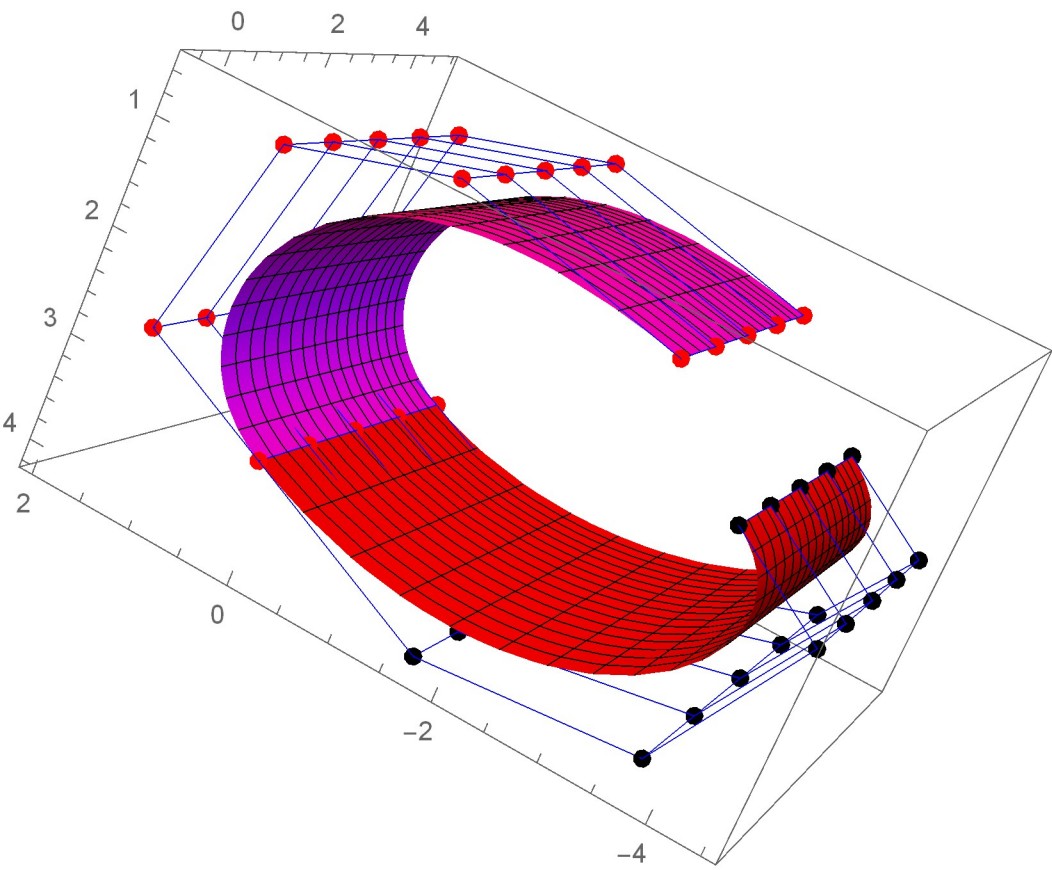

**Fig 31. The quantic Bézier- like surface for** $f = 0.9$, $\varsigma = \varsigma^* = 1.6$, $\zeta = \zeta^* = 1.6$.

It implies that:

$$\sum_{f=0}^{r} \breve{u}_{f,r}(\tilde{w}, \varsigma, \zeta)[s1(2\zeta 1 + s1 - 5)(D_{f,s1} - 2D_{f,s1-1} + D_{f,s1-2}) + 2(-6 + 2\zeta 1$$

$$+\varsigma 1)(D_{f,s1-1} - D_{f,s1-2})] = z^2 \sum_{g=0}^{r} \breve{u}_{f,r}(\tilde{w}, \varsigma^*, \zeta^*)[s2(2\varsigma 1^* + s2 - 5)(D_{0,g} - 2D_{1,g} \tag{50}$$

$$+D_{2,g}) + 2(-6 + 2\varsigma 1^* + \zeta 1^*)(D_{1,g} - D_{2,g})].$$

If $\varsigma = \varsigma^*$ and $\zeta = \zeta^*$, then

$$\begin{aligned} & s1(2\zeta 1 + s1 - 5)(D_{f,s1} - 2D_{f,s1-1} + D_{f,s1-2}) + 2(-6 + 2\zeta 1 + \varsigma 1)(D_{f,s1-1} - D_{f,s1-2}) \\ = \ & z^2[s2(2\varsigma^* + s2 - 5)(D_{f,0} - 2D_{f,1} + D_{f,2}) + 2(-6 + 2\varsigma^* + \zeta^*)(D_{f,1} - D_{f,2})]. \end{aligned} \tag{51}$$

Finally, if two surfaces $W_{n1,s1}(\tilde{w}, \tilde{w1}; \varsigma, \varsigma 1, \zeta, \zeta 1)$ and $W1_{n2,s2}(\tilde{w}, \tilde{w1}; \varsigma^*, \varsigma 1^*, \zeta^*, \zeta 1^*)$ satisfy Eqs (43) and (48) and (51), then both surfaces are connected by $G^2$ continuity in $\tilde{w}$ and $\tilde{w1}$ directions.

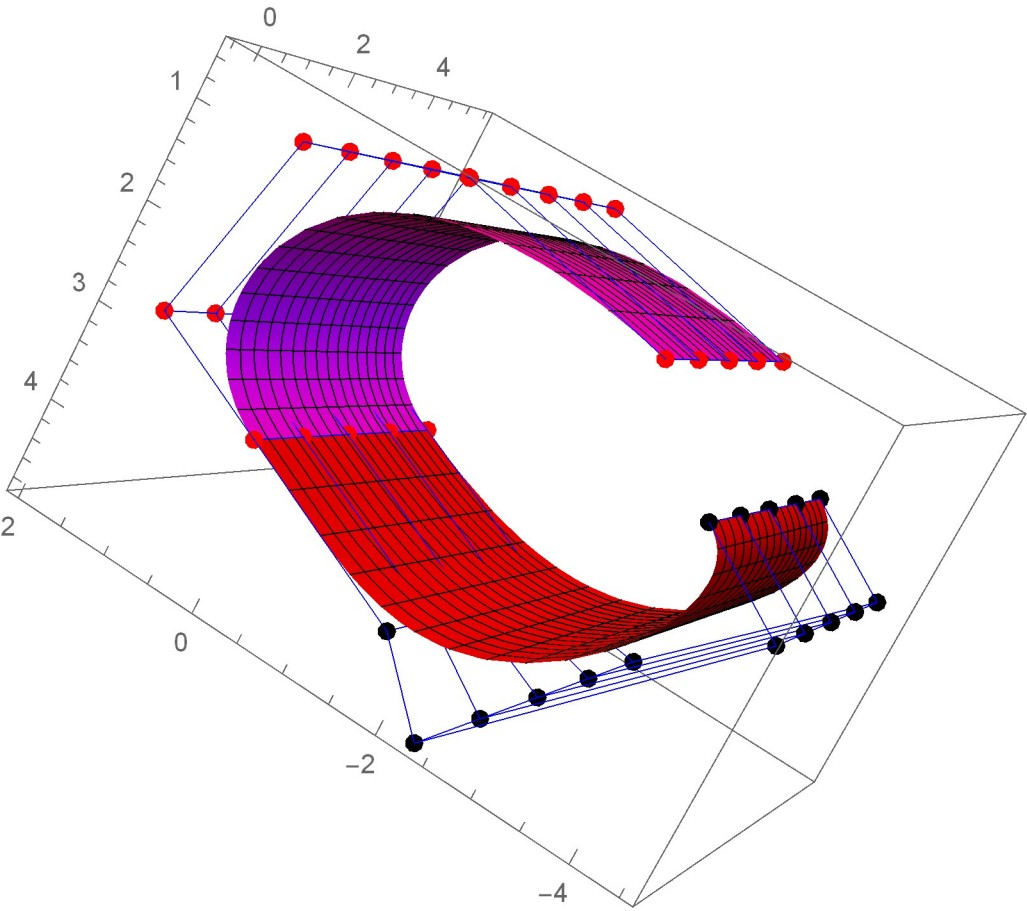

**Fig 32. The quantic Bézier- like surface for $f = 1.2$, $\varsigma = \varsigma^* = 1.6$, $\zeta = \zeta^* = 1.6$.**

## 3.4 Continuity in $\tilde{w}1$ direction

Similarly, the $G^1$ and $G^2$ continuity can be obtained in the direction of $\tilde{w}1$ for two gBS as the continuity in the direction of $\tilde{w}$ for two gBS, which is discussed in detail in Section 3.2. For two Bézier-like surfaces, the $G^2$ continuity is also explored in the $\tilde{w}$ direction. The continuity of $G1$ among Bézier-like surfaces areas for $\zeta = \zeta^*$ $\varsigma = \varsigma^*$ has mutual control points $r1 = r2 = r$, and variations in $D_{f,s1-1}$, $D_{f,s1}$, $D1_{f,1}$. The $G^2$ continuity of two areas among Bézier-like surfaces considered for $\varsigma = \varsigma1^*$ and $\zeta = \zeta1^*$ in $\tilde{w}1$ and $\tilde{w}$ directions with $r1 = s2$ common control points, and the control points $D_{f,s1-1}$, $D_{f,s1}$ (or $D1_{0,f}$), $D1_{1,g}$ are taken in varied order.

Figs 21 and 22 are modeled with the help of $G^2$ surface continuity constraint. Fig 21 shows the two cubic Bézier-like surfaces, which are connected with $G^2$ surface continuity constraints. Smooth connection between these two cubic surfaces shows that our continuity constraints are valid. We can alter the figure by just changing the value of shape parameters and scaling factor. Similarly, Fig 22 depicts the two quantic Bézier-like surfaces, which are linked by $G^2$ surface continuity constraints.

By changing the value of shape parameters and scaling factor, the surface and control net can be altered by our choice. When the shape parameters have a fixed value and the value of scaling factor varies then the control points $D1_{1,i}$ exhibit a tendency to move either closer to or

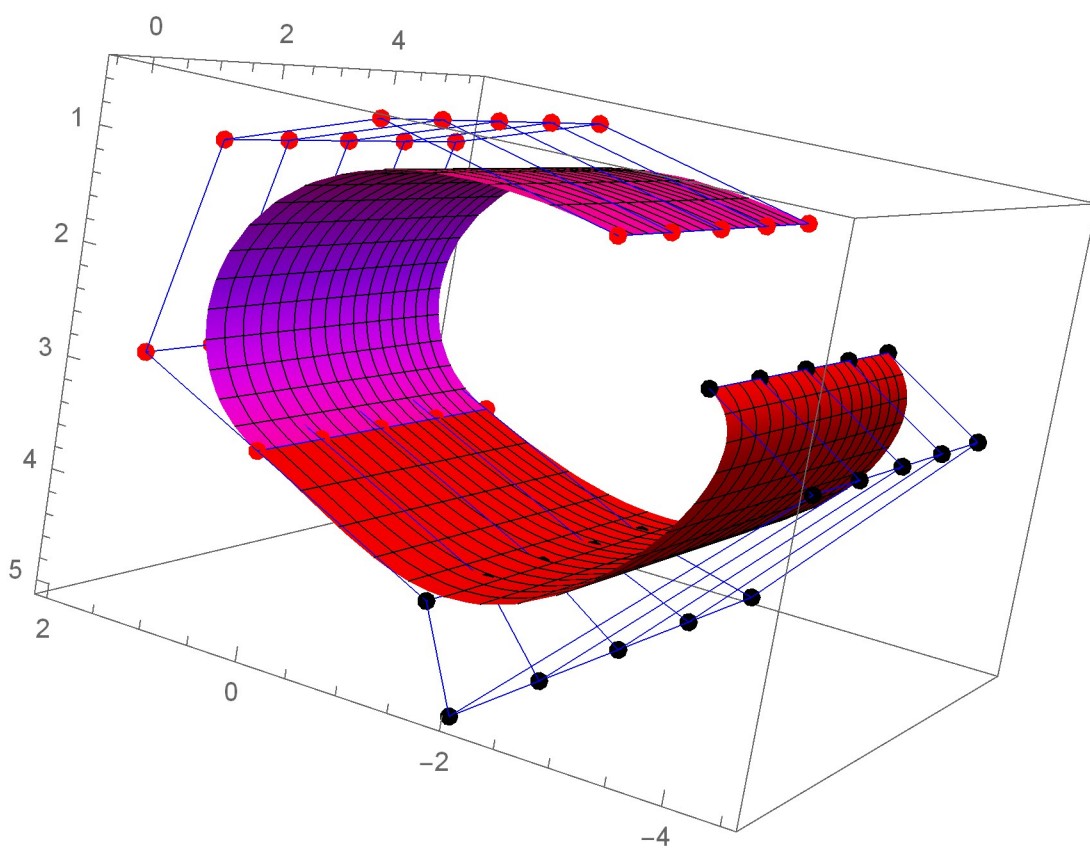

**Fig 33. The quantic Bézier- like surface for $f = 1.3$, $\varsigma = \varsigma^* = 1.6$, $\zeta = \zeta^* = 1.6$.**

farther away from the control points $D1_{2,i}$, depending on whether the value of scaling factor $f$ raises or reduces. Similarly, when the scaling factor $f$ is kept constant and value of the shape parameters is increased or decreased, the surface demonstrates a tendency to move either closer to or farther away from the control polygon (control net). The present investigation demonstrates that the piecewise Bézier-Like surface exhibits smoothness and continuity at the joints, hence enhancing its efficacy in addressing challenges related to engineering appearance design through the modification of surface position and shape.

The effect of shape parameters and scaling factor can be shown in the few surface figures. Figs 23–26 depict the $G^2$ surface continuity of two Bézier-Like surfaces for different values of shape parameters. Note that from Figs 23–26, when the values of shape parameters are increases and the scaling factor remains constant at 1.5 then the surface moves closer to the control net. On the other hand as in the Figs 27–30 the scaling factor is at constant value 0.9 and the value of shape parameter is increased then the surface moves away from the control net. Figs 31–34 has the same value of shape parameters but the value of scaling factor varies. The control points $D1_{2,i}$ move closer to the control points $D1_{1,i}$ as the value of scaling factor $f$ increased. By varying the value of shape parameters and scaling factors the surface can be modeled according to our choice which is very useful to model the complex surfaces in the field of engineering.

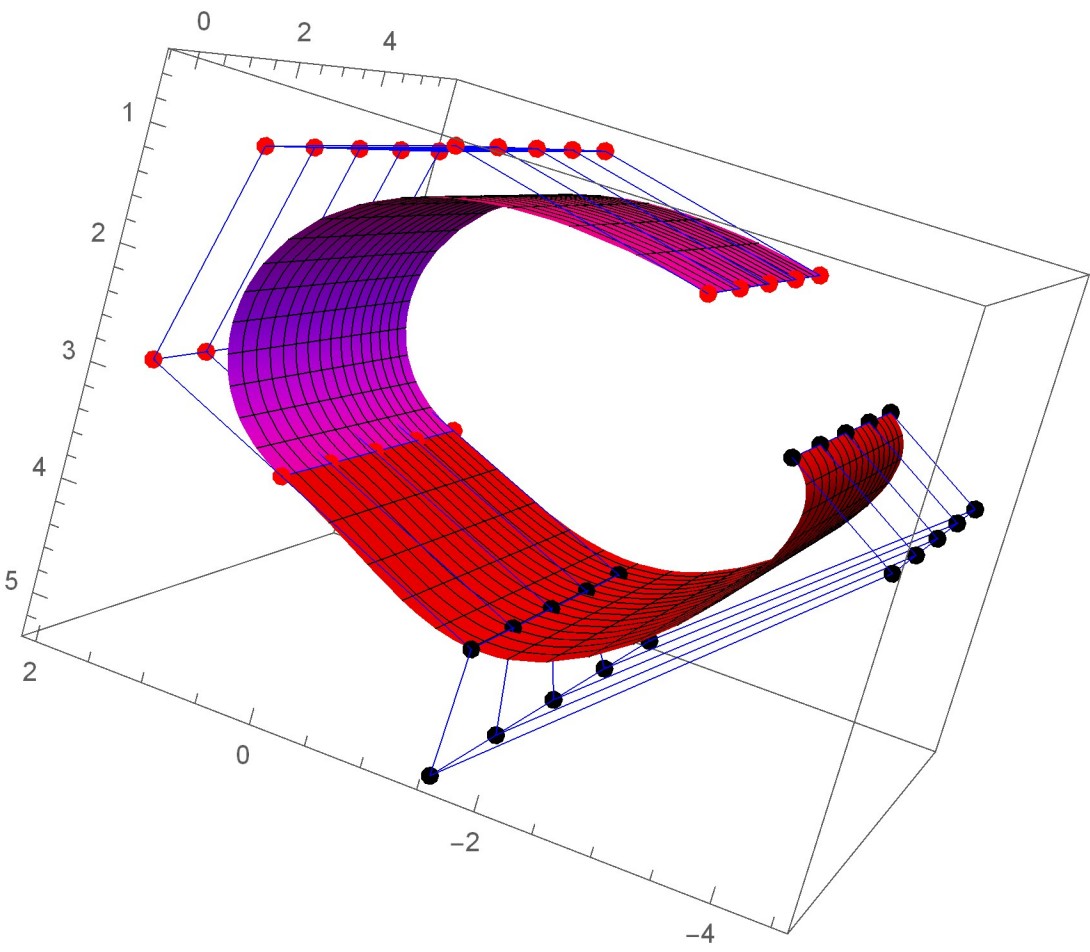

**Fig 34. The quantic Bézier- like surface for** $f = 1.6$, $\varsigma = \varsigma^* = 1.6$, $\zeta = \zeta^* = 1.6$.

## 4 Conclusions

Bézier curves are versatile mathematical functions commonly used to represent smooth curves and shapes in digital environments. Their uses extend to a wide range of applications, including editing images, creating visual compositions, and designing fonts with precise curves and contours. Bézier curves offer a powerful tool for creating intricate and visually appealing designs in order to increase the number of control points, allowing for greater control and precision compared to simple straight lines. Any curved shape might be achieved by increasing the number of control points. However, simply using a gBC is insufficient when discussing the modeling of complicated figures and font design. In order to address this problem, this research derived the parametric and geometric conditions of degree three ($C^3$ and $G^3$ continuity) between any two gBC. These special continuity preserving curves are utilized in CAD/CAM and have shape-controlling parameters. Moreover, we proposed continuity constraints ($G^1$ and $G^2$ continuity) between two gBS with different shape parameters to address the issue of modeling and designing surfaces. Using the different values of shape parameters, we can alter the shape of surfaces by our choice, which is very helpful in modeling some complex surfaces in engineering and some other fields as well. This integration gains more control and adjustability over the curves and surfaces through the addition of the shape parameter. Our

research is significant and useful since our proposal enables the construction of more easily-realized complex curves and surfaces using computer. These gB-Like functions will find application in quantum engineering and post quantum calculus frames in the future, approximation theory and CAGD researchers may find them interesting. Furthermore, it is suggested to extend the application of gBS to the creation of developable surfaces. The implementation of shape parameters in the creation of developable surfaces is anticipated to offer more versatility and adaptability in relation to the size of surface patches.

## Acknowledgments

The authors are grateful to anonymous referees for their valuable suggestions, which significantly improved this manuscript.

## Author Contributions

**Investigation:** Madiha Shafiq.

**Methodology:** Moavia Ameer, Muhammad Abbas, Madiha Shafiq, Tahir Nazir, Asnake Birhanu.

**Project administration:** Muhammad Abbas.

**Software:** Moavia Ameer, Madiha Shafiq.

**Supervision:** Muhammad Abbas, Tahir Nazir.

**Validation:** Madiha Shafiq.

**Visualization:** Moavia Ameer, Muhammad Abbas, Madiha Shafiq, Tahir Nazir, Asnake Birhanu.

**Writing – original draft:** Moavia Ameer, Muhammad Abbas, Tahir Nazir, Asnake Birhanu.

**Writing – review & editing:** Moavia Ameer, Muhammad Abbas, Madiha Shafiq.

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
