## [Decision Letter · Decision Letter 0]

5 Feb 2024

PONE-D-23-44182Generalized Bézier-like model and its applications to curve and surface modelingPLOS ONE

Dear Dr. Birhanu,

Thank you for submitting your manuscript to PLOS ONE. After careful consideration, we feel that it has merit but does not fully meet PLOS ONE’s publication criteria as it currently stands. Therefore, we invite you to submit a revised version of the manuscript that addresses the points raised during the review process.

The manuscript have many flaws need to be addressed are as follows:** 1. Novelty of the work is missing****2. In depth literature review need to added****3. Mathematical expression have many error****4. Lacking the presentation of Result and discussion section****5. Need to improve the conclusion section**

We look forward to receiving your revised manuscript.

Kind regards,

Sameer Sheshrao Gajghate, PhD

Academic Editor

PLOS ONE

Reviewers' comments:

Reviewer's Responses to Questions

**Comments to the Author**

1. Is the manuscript technically sound, and do the data support the conclusions?

Reviewer #1: Yes

Reviewer #2: Yes

Reviewer #3: Partly

2. Has the statistical analysis been performed appropriately and rigorously? 

Reviewer #1: N/A

Reviewer #2: Yes

Reviewer #3: N/A

3. Have the authors made all data underlying the findings in their manuscript fully available?

Reviewer #1: Yes

Reviewer #2: Yes

Reviewer #3: No

4. Is the manuscript presented in an intelligible fashion and written in standard English?

Reviewer #1: Yes

Reviewer #2: Yes

Reviewer #3: Yes

5. Review Comments to the Author

Reviewer #1: Manuscript ID: PONE-D-23-44182

The manuscript entitled "Generalized Bézier-like model and its applications to curve and surface modelling” proposes C^3 and G^3 continuity for curves, and G^1 and G^2 continuity for surface. This study is an extension of [1] and [2]. Hence, this work is the followed-up work. Overall, the paper has interesting results and significant contribution. However, the current of this work cannot be published until the following problems and remarks are addressed.

Minor comments:

1. There is typo in Equation (2.2) and (2.4) it should be U(F) since the author denoted the symbol for parametric curve.

2. Please mention that u and v are the x and y components of U(F), respectively for Equation (2.3) and (2.5).

3. In Equation (2.10), there is typo, it should be 0 ≤w, w1 ≤1.

4. Make sure the partition of each section is well organized. For example, Section 2.6 is for continuity of curve and surfaces but only discuss for curve only while Section 3 discuss for surface continuity.

5. In Section 3.2 first sentence, it should be G^0 continuity not G^1.

6. In Section 3.2 last sentence, it should G^2 continuity in w direction not t.

7. On page 13, last paragraph, please recheck which said figures.

Please check the manuscript for typos, grammatical mistakes, and undefined symbols.

Major remarks/comments:

1. The abstract should answer the following: What is the motivation of this study? What is the current limitation of the existing method? What is the advantage of the proposed method compared to the existing method?

2. Lack of literature review regarding G^1/G^2 continuity of Bezier-like surface. Consider the following work:

a) Hu, G., Cao, H., Wang, X., & Qin, X. (2017). G^2 continuity conditions for generalized Bézier-like surfaces with multiple shape parameters. Journal of Inequalities and Applications, 2017, 1-17.

b) Syed Ahmad Aidil Adha Said Mad Zain, Md Yushalify Misro. A novel technique on flexibility and adjustability of generalized fractional Bézier surface patch[J]. AIMS Mathematics, 2023, 8(1): 550-589. doi: 10.3934/math.2023026

3. Introduction has unclear motivation. What is the research gap? Next, what is the novelty of this research?

4. In Section 2.4 and 2.5, there is a lack of discussion regarding the geometric effect of shape parameters to the curves and surfaces. Please give details regarding the geometric effect of shape parameters, for example what will happen if the first shape parameter is increased while the other parameter is decreased and vice versa. What would happen if both increased/decreased?

5. In Section 2.6 paragraph 1, please elaborate regarding the reason C^3/G^3 continuity is smoother than C^2/G^2. What is the significant advantage of C^3/G^3 continuity compared to C^2/G^2 in terms of visualization? Please include the curvature comb or at least curvature plot to distinguish between C^2/G^2 continuity and C^3/G^3 continuity. You may refer to these references for curvature comb:

a) Farin, G. (2016). Curvature combs and curvature plots. Computer-Aided Design, 80, 6-8.

b) S.A.A.A. Said Mad Zain, M.Y. Misro, K.T. Miura, Curve fitting using generalized fractional Bézier curve, Computer-Aided Des. Appl. 20 (2023) 350–363.

6. In Section 2.6, it is preferable to include figures of C^2/G^2 continuity of gB-curves for comparison between C^2/G^2 and C^3/G^3 continuity.

7. In Section 3.2, 3.3 and 3.4 it is better to write the conditions and equations of G^2 continuity of surface in a proper theorem. It is quite weird because for Section 2.6 there is proper theorem for curve continuity. You can refer to reference in point (2).

8. Please recheck Eqn (3.9) and (3.10).

9. Please change title section 3.5 since it is confusing with section 3.3. Rewrite as the procedure and examples of constructing G2 surfaces using theorems.

10. Rewrite the conclusion by considering the motivation and advantages of the proposed method. Please also include the limitations of the proposed method.

If the author can address the comments, then I will recommend this manuscript for publication.

Reviewer #2: This paper is about the explanation of a novel Bezier-like curve definition. Apart from a few grammar errors, the text is well-written. However, it must be checked (again) by a native English-speaker. They used Bernstein-like functions as the basis of their construction. I didn't check all the equations and calculations of this paper. It looks like doable anyways. Therefore, I am more concerned with the idea and the novelty of the method this study proposes. I find them original and see it as positive contributions to the field.

Reviewer #3: This article extends the previous work of Amber et al. (2022a, 2022b) by introducing Generalised Bezier-like C^3 and G^3 curve and surface design.

Overall acceptable, but there are many typos exist in the present version. Below are comments to improve the paper.

1. Abstract: there is no evident in the article to support the proposed method is better than the standard Bézier curves and surfaces. This is the major issue in this article, show clearly how the proposed curve/surface better than traditional Bézier model.

2. pg.2: Before writing the short form, make sure to write the full form, e.g "H-BS", "GHT-BS" etc.

3. pg.2: the contribution of this paper is the continuity extension, the rest of the points are either redundant or has been established in Amber et al. (2022a, 2022b)

4. eq.(2.1) should;d be written as M =(u,v); this notation is inline with the rest of equations.

5. pg.3: U(Φ) should be written as M(Φ).

6. what is gB-like? state the full form first.

7. pg.4: define C_f stated in eq.(2.8)

8. eq.(2.10) typo: "0<=w,w1<=1", not "0<=w,w1<=r"

9. replace "talk" with "discuss".

10. Theorem 1: state for "parametric continuity C^3", the rest has been established in Amber et al. (2022a, 2022b).

11. C^f or C_f? be consistent.

12. Theorem 2: state for "parametric continuity G^3", the rest has been established in Amber et al. (2022a, 2022b).

13. what is φ?

14. Typo: the Fig.1, not Fig.2

15. State all the details to produce Figure 1. This applies for the rest of figures too, hence increasing reproducibility.

16. equatios stated after eq.(3.10) has typo: replace d(z) with d(w).

17. typo in eq.(3.12), m1?

18. typo in equation citing: Substituting (3.12) and (3.13) into (3.25)?

19. pg.13: typo Fig.3(a)....Fig.3(b)

6. PLOS authors have the option to publish the peer review history of their article (what does this mean?). If published, this will include your full peer review and any attached files.

Reviewer #1: No

Reviewer #2: No

Reviewer #3: No

---

## [Author Response · Author response to Decision Letter 0]

1 Apr 2024

Dear professor,

Please see the attached response file.

Thanks

---

## [Decision Letter · Decision Letter 1]

15 Apr 2024

Generalized Bézier-like model and its applications to curve and surface modeling

PONE-D-23-44182R1

Dear Dr. Asnake Birhanu,

We’re pleased to inform you that your manuscript has been judged scientifically suitable for publication and will be formally accepted for publication once it meets all outstanding technical requirements.

Kind regards,

Sameer Sheshrao Gajghate, PhD

Academic Editor

PLOS ONE

Additional Editor Comments (optional):

Reviewers' comments:

Reviewer's Responses to Questions

**Comments to the Author**

1. If the authors have adequately addressed your comments raised in a previous round of review and you feel that this manuscript is now acceptable for publication, you may indicate that here to bypass the “Comments to the Author” section, enter your conflict of interest statement in the “Confidential to Editor” section, and submit your "Accept" recommendation.

Reviewer #1: All comments have been addressed

Reviewer #3: All comments have been addressed

2. Is the manuscript technically sound, and do the data support the conclusions?

Reviewer #1: Yes

Reviewer #3: Partly

3. Has the statistical analysis been performed appropriately and rigorously? 

Reviewer #1: N/A

Reviewer #3: N/A

4. Have the authors made all data underlying the findings in their manuscript fully available?

Reviewer #1: Yes

Reviewer #3: Yes

5. Is the manuscript presented in an intelligible fashion and written in standard English?

Reviewer #1: Yes

Reviewer #3: Yes

6. Review Comments to the Author

Reviewer #1: The manuscript was improved according to the reviewer's comments and suggestions. Hence, I recommend the manuscript to be accepted for publication.

Reviewer #3: (No Response)

7. PLOS authors have the option to publish the peer review history of their article (what does this mean?). If published, this will include your full peer review and any attached files.

Reviewer #1: No

Reviewer #3: No

---

## [Editor Report · Acceptance letter]

22 May 2024

PONE-D-23-44182R1 

PLOS ONE

Dear Dr. Birhanu, 

I'm pleased to inform you that your manuscript has been deemed suitable for publication in PLOS ONE. Congratulations! Your manuscript is now being handed over to our production team.

Kind regards, 

on behalf of

Dr. Sameer Sheshrao Gajghate 

Academic Editor

PLOS ONE